# Oxidative Stress and Redox Imbalance: Common Mechanisms in Cancer Stem Cells and Neurodegenerative Diseases

**DOI:** 10.3390/cells14070511

**Published:** 2025-03-29

**Authors:** Nikhil Raj Selvaraj, Durga Nandan, Bipin G. Nair, Vipin A. Nair, Parvathy Venugopal, Rajaguru Aradhya

**Affiliations:** School of Biotechnology, Amrita Vishwa Vidyapeetham, Kollam 690525, Kerala, India; nikhilraj.ceib@gmail.com (N.R.S.); bhilae@gmail.com (D.N.); bipin@am.amrita.edu (B.G.N.); vipinanair@am.amrita.edu (V.A.N.)

**Keywords:** oxidative stress, redox imbalance, cancer stem cells, neurodegenerative diseases, reactive oxygen species, mitochondrial dysfunction, oxidative phosphorylation, ferroptosis, autophagy, antioxidant

## Abstract

Oxidative stress (OS) is an established hallmark of cancer and neurodegenerative disorders (NDDs), which contributes to genomic instability and neuronal loss. This review explores the contrasting role of OS in cancer stem cells (CSCs) and NDDs. Elevated levels of reactive oxygen species (ROS) contribute to genomic instability and promote tumor initiation and progression in CSCs, while in NDDs such as Alzheimer’s and Parkinson’s disease, OS accelerates neuronal death and impairs cellular repair mechanisms. Both scenarios involve disruption of the delicate balance between pro-oxidant and antioxidant systems, which leads to chronic oxidative stress. Notably, CSCs and neurons display alterations in redox-sensitive signaling pathways, including Nrf2 and NF-κB, which influence cell survival, proliferation, and differentiation. Mitochondrial dynamics further illustrate these differences: enhanced function in CSCs supports adaptability and survival, whereas impairments in neurons heighten vulnerability. Understanding these common mechanisms of OS-induced redox imbalance may provide insights for developing interventions, addressing aging hallmarks, and potentially mitigating or preventing both cancer and NDDs.

## 1. Introduction

Oxidative stress (OS), which results from an imbalance in redox states, plays a crucial role in the pathophysiology of various diseases, including cancer and neurodegenerative disorders (NDDs). This imbalance leads to cellular damage, disrupted signaling pathways, and alterations in metabolic homeostasis, thereby driving disease progression [1].

Cancer stem cells (CSCs) are a unique subpopulation of tumor cells with self-renewal capabilities, which are responsible for tumor initiation, progression, and metastasis [2]. These cells have developed adaptive mechanisms to counteract free radical toxicity, including activation of redox-sensitive transcription factors and increased expression of antioxidant enzymes and anti-apoptotic proteins [3]. Similarly, NDDs exhibit distinct OS patterns such as oxidative damage, mitochondrial dysfunction, and misfolded protein accumulation [4]. In Alzheimer’s disease, for instance, increased protein and lipid oxidation in the brain strongly correlates with cognitive decline and neuronal death [5].

The molecular mechanisms linking OS in CSCs and NDDs share several key features, with mitochondrial dysfunction playing a central role. Affected neurons and CSCs exhibit altered mitochondrial activity, increased reactive oxygen species (ROS) production, and impaired energy metabolism [6]. Both conditions disrupt antioxidant defense systems, with CSCs often showing elevated levels of superoxide dismutase (SOD2) for neutralizing OS, while neurons from Alzheimer’s patients and animal models display increased ROS production, DNA damage, and impaired oxidative phosphorylation [7,8].

Common disruptions in these conditions, such as impaired autophagy, protein homeostasis, genomic instability, inflammatory responses, alterations in the cellular microenvironment, and oxidative stress-driven cellular senescence, further exacerbate disease progression in both cases [9]. Recognizing these shared pathways presents opportunities for developing therapeutic strategies that target OS and redox imbalances. Potential approaches include tailored antioxidant therapies, mitochondrial-targeted drugs, and NRF2 pathway modulators that are adapted to the specific needs of CSCs or neurons [10,11,12]. For example, enhancing autophagy could help clear toxic protein aggregates in NDDs, whereas inhibiting it may weaken CSCs and increase their susceptibility to treatment [13,14]. Similarly, metabolic interventions could exploit vulnerabilities in CSCs while protecting neurons against oxidative damage [15].

This review offers a novel perspective by integrating insights from CSC research and NDD studies, highlighting shared mechanisms of oxidative stress and redox imbalance. It emphasizes the dual role of OS as a “double-edged sword” in CSC development and progression while also being central to NDDs like Alzheimer’s and Parkinson’s. The review identifies common mechanisms between CSCs and NDDs, including mitochondrial dysfunction, altered redox signaling, and shared pathways like NRF2 signaling. A key insight is the focus on the interplay between oxidative stress and cellular adaptation mechanisms. It explores how CSCs and neurons in neurodegenerative conditions develop complex systems to adapt to high ROS environments. This comparative analysis offers novel perspectives on cellular resilience and vulnerability across different disease contexts, potentially opening new avenues for therapeutic interventions in both cancer and NDDs. In conclusion, the shared role of OS in CSCs and NDDs underscores the importance of maintaining redox balance in cellular health. This integrated approach provides a comprehensive understanding of how redox dysregulation impacts both CSC and neurodegeneration through common pathways, offering potential opportunities for innovative therapeutic strategies.

## 2. Oxidative Stress and Redox Imbalance: An Overview

OS occurs when ROS and reactive nitrogen species (RNS) overwhelm the antioxidant capacity of the body, causing damage to lipids, proteins, and DNA (Figure 1) [16]. Although moderate levels of ROS and RNS are essential for cellular signaling and homeostasis [17], disruption of this balance can lead to various pathological conditions such as cancer, cardiovascular disorders, and neurodegenerative conditions [18,19].

Interestingly, while excessive ROS can cause oxidative damage, low levels of ROS play important roles in cell signaling and physiological processes. This dual nature of ROS is referred to as redox biology, wherein a small increase in ROS level can activate signaling pathways to initiate biological processes, while high levels result in OS and cellular damage [20]. Redox imbalance can present as oxidative or reductive stress, both of which are potentially detrimental to cellular functions [21]. The cellular redox environment is maintained by enzymes and antioxidants through constant metabolic energy input [22]. Understanding the delicate balance between cellular redox systems and their compartment-specific concentrations is imperative [22].

### 2.1. Mechanisms of Redox Regulation

Redox regulation is a physiological process involving the controlled production and decay of redox-active molecules such as superoxide, hydrogen peroxide, nitric oxide, and hydrogen sulfide [23]. These molecules are produced by enzymes such as NADPH oxidases, superoxide dismutases, and nitric oxide synthases. They interact with target proteins and modify cysteine residues in various signaling cascades, which play essential roles in cellular processes. The thioredoxin family proteins are key regulators of this process, controlling the formation and removal of oxidative modifications through thiol reduction and oxidation [24]. This system influences numerous cellular functions, including the inflammatory and immune responses [23].

Redox regulation affects several important signaling pathways (Table 1):NF-κB pathway: A master regulator of inflammatory responses that is highly sensitive to redox changes. It forms specific signaling complexes that regulate target gene expression during acute inflammation [25].MAPK pathway: Redox-sensitive and involved in cellular responses to OS [26].Nrf2-Keap1 pathway: Central to cellular defense against oxidative and electrophilic insults [27]. Nrf2 binds to the Antioxidant Response Element (ARE) to induce the expression of antioxidant and detoxifying enzymes [27]. Keap1 negatively regulates Nrf2 by targeting it for degradation. Disrupting the Keap1-Nrf2 interaction can enhance the antioxidant capacity of the brain and protect against OS and neuroinflammation [28].

**Table 1 cells-14-00511-t001:** Context-dependent roles of NF-κB, MAPK, and KEAP1-NRF2 pathways in CSCs vs. NDDs.

Pathway	Cancer Stem Cells (CSCs)	Neurons
MAPK Pathway (Mitogen-Activated Protein Kinase)	ROS activates ERK, JNK, and p38 MAPK, promoting self-renewal, survival, and proliferation [29]. Moderate ROS supports CSC stemness, while excessive ROS can induce differentiation or apoptosis [30].	ROS-dependent activation of JNK and p38 MAPK is linked to neurodegeneration and apoptosis [31]. ERK signaling regulates synaptic plasticity and memory but is impaired in oxidative stress conditions [32]. Chronic p38 MAPK activation contributes to neuroinflammation and neuronal loss [33].
NF-κB Pathway (Nuclear Factor Kappa B)	Constitutively active NF-κB promotes CSC survival, immune evasion, and therapy resistance [34]. ROS-induced NF-κB activation enhances anti-apoptotic genes [35] Cross-talk with Notch and Wnt pathways sustains stemness and tumor progression [36].	NF-κB activation by ROS contributes to neuroinflammation and neurodegenerative diseases [37]. Upregulation of pro-inflammatory cytokines (IL-6) accelerates neuronal damage [38]. Chronic NF-κB activation in glial cells exacerbates oxidative stress and neurotoxicity [38].
Nrf2 Pathway (Nuclear Factor Erythroid 2-Related Factor 2)	Highly active in CSCs, maintaining redox homeostasis and therapy resistance [39] Nrf2-driven upregulation of antioxidant enzymes (GSH, SOD, CAT) neutralizes ROS [40]. Promotes metabolic adaptation via pentose phosphate pathway (PPP) for NADPH production [41].	Active under physiological conditions, but declines with age and neurodegeneration [42]. Reduced Nrf2 activity in Alzheimer’s and Parkinson’s leads to oxidative damage [43]. Impaired Nrf2 signaling contributes to mitochondrial dysfunction and neuronal apoptosis [44].

These redox-sensitive pathways form a complex network that regulates the cellular responses to OS and inflammation [45]. The Nrf2-Keap1 pathway primarily mediates antioxidant defenses, whereas the NF-κB and MAPK pathways are involved in both pro-inflammatory and stress responses [46].

### 2.2. Consequences of OS and Redox Imbalance

OS can cause damage to major biomolecules, including lipids, proteins, carbohydrates, and nucleic acids [47]. This damage can lead to cellular dysfunction, premature cell death, and inflammation [48]. In the immune system, increased ROS levels can lead to either proliferation or programmed cell death in lymphocytes, with HIV-1-infected individuals exhibiting increased susceptibility to apoptosis [49]. Ferroptosis, marked by the buildup of iron ions and lipid peroxides, is intricately associated with OS and has been implicated in numerous pathological conditions [50]. OS-induced ferroptosis in cardiomyocytes involves complex mechanisms, including GSH depletion and GPX4 degradation [51]. Additionally, OS influences cell cycle dysregulation and proliferation, as demonstrated by the effects of 25-hydroxycholesterol on extravillous trophoblasts [52]. It also plays a crucial role in the pathophysiology of genetic conditions, such as Down syndrome and Williams-Beuren syndrome, contributing to early aging, dementia, autoimmunity, and chronic inflammation [53,54].

### 2.3. Dual Roles of OS

ROS exhibit a complex dual nature in biological systems, acting as both beneficial signaling molecules and potential damaging agents [55]. At low concentrations, ROS function as essential messengers in cellular signaling pathways, promoting health and potentially extending lifespan through mitohormesis [56]. These beneficial effects include roles in cellular responses to hypoxia, defense against infectious agents, and various cellular processes such as differentiation, proliferation, and immune responses [57]. The beneficial effects of low-level ROS signaling are evident in various contexts. For instance, calorie restriction, hypoxia, temperature stress, and physical activity can trigger adaptive responses through moderate ROS production. In exercise physiology, moderate levels of free radicals induce the body’s antioxidant defenses and improve muscle adaptation [58]. Additionally, low levels of ROS can induce a mitogenic response.

However, as ROS concentrations increase, their effects transition from beneficial to harmful. High concentrations of ROS lead to programmed cell death [57] and can cause significant damage to cell structures, including lipids, membranes, proteins, and nucleic acids, a phenomenon termed OS. To counteract these harmful effects, cells employ an antioxidant defense system comprising both nonenzymatic antioxidants and antioxidant enzymes [59]. Despite these protective mechanisms, oxidative damage accumulates over time, potentially contributing to the development of age-dependent diseases.

The dual nature of ROS is particularly evident in CSCs and NDDs. In CSCs, ROS can act as signaling molecules to maintain stemness and promote tumor progression [60] while also potentially inducing differentiation and cell death at higher levels [61]. While in NDDs, ROS contribute to neuronal damage. The relationship between ROS and CSCs is complex, as ROS can both promote and inhibit CSC functions depending on the context and level of OS [62]. Some studies suggest that CSCs maintain low ROS levels for self-renewal and survival [39], while others indicate that elevated mitochondrial ROS in certain CSC populations can drive metastasis formation [63].

This duality presents both challenges and opportunities for therapeutic approaches. In cancer treatment, strategies aimed at elevating ROS levels in CSCs may be effective in eliminating these therapy-resistant cells [64]. Conversely, for NDDs, approaches focusing on reducing ROS production or enhancing antioxidant defenses, such as the use of flavonoids, polyphenols may help delay disease progression [65]. However, the complex role of ROS in cellular processes necessitates careful consideration of potential side effects and the development of targeted approaches to modulate ROS levels in specific cell populations.

### 2.4. Consequences of Oxidative Stress: Cancer Stem Cells and Neurodegeneration

OS plays a crucial role in both cancer and NDDs and presents challenges and opportunities for therapeutic interventions. Increased OS can selectively kill cancer cells, including CSCs, by generating ROS [66]. However, CSCs may develop adaptive mechanisms to counteract these effects, contributing to resistance to therapy. Interestingly, CSCs maintain lower ROS levels than non-CSCs, which may be critical for their survival and function [3]. This characteristic presents a potential therapeutic target, as ROS-generating drugs could potentially eradicate drug-resistant CSCs [67].

OS contributes to neuronal dysfunction, synaptic damage, and neurodegeneration in NDDs [68]. In Alzheimer’s disease (AD), OS precedes amyloid plaque and neurofibrillary tangle formation [69]. Mutations in amyloid beta-protein precursor and tau genes can induce mitochondrial dysfunction and OS [70]. OS is linked to α-synuclein aggregation in Lewy bodies in Parkinson’s disease (PD) [71]. Environmental factors, such as exposure to neurotoxic metals and pesticides, contribute to the development of both AD and PD through OS mechanisms [72].

The shared pathways of OS in cancer and NDDs suggest the potential of targeting redox imbalances in both conditions [73]. For instance, the endocannabinoid system modulates redox-state-dependent cell death and inflammation in NDDs [74], whereas cancer cells activate redox-sensitive transcription factors as an adaptive response [75]. Understanding these shared pathways may lead to the development of novel therapeutic strategies. However, the complex role of OS necessitates careful consideration when developing targeted interventions.

The following sections will examine in greater detail the specific mechanisms underlying this redox-driven pathology in CSCs and NDDs.

## 3. Oxidative Stress in Cancer Stem Cells

Despite advancements in oncology, challenges such as therapy resistance, relapse, and metastasis persist, with cellular heterogeneity and the tumor microenvironment significantly influencing clinical outcomes [76]. CSCs, a small subpopulation of tumors, play a crucial role in these challenges [77]. CSCs possess unique properties, including self-renewal, high proliferative capacity, metastatic potential, and resistance to conventional therapies. These features enable CSCs to evade treatment by actively dividing tumor cells and persist within the tumor niche [78].

CSCs reside in specific niches within the tumor microenvironment, particularly in perivascular and hypoxic regions [79]. Resistance to chemotherapy and radiation is partly due to the expression of drug efflux pumps, such as ABCG2, and the activation of growth factor pathways [80]. Paradoxically, CSCs can benefit from anti-angiogenesis therapy owing to increased tumor hypoxia [81]. Their immunomodulatory traits also allow them to evade detection by the immune system. Targeting CSCs without harming normal stem cells highlights the need for therapies that can selectively disrupt CSC redox homeostasis [82]. This approach can prevent tumor recurrence and combat aggressive treatment-resistant forms of cancer. Consequently, understanding how CSCs regulate OS and maintain redox homeostasis has become a major focus of CSC research [83].

Cancer cells, particularly CSCs, exhibit elevated ROS levels compared to normal cells. For instance, leukemia cells from chronic lymphocytic leukemia patients show increased ROS production compared to normal lymphocytes [84]. Interestingly, glioblastoma stem cells (GSCs) deviate from this pattern by maintaining lower ROS levels [15]. This characteristic contributes to their self-renewal capacity and resistance to radiotherapy [85]. GSCs achieve this by regulating mitochondrial ROS production through the stabilization of peroxiredoxin3 (PRDX3), a mitochondrion-specific peroxidase [86]. This mechanism highlights the complex relationship between ROS levels and cancer cell behavior, particularly in stem-like populations. CSCs possess a unique ability to maintain redox homeostasis [87], enabling them to survive conventional cancer treatments that target non-stem cancer cells [88]. The generation and control of ROS are central to CSC survival, as these cells have evolved mechanisms to modulate ROS production and detoxification, allowing them to thrive under adverse conditions.

While conventional cancer therapies like chemotherapy and radiotherapy effectively induce ROS-mediated cell death in bulk tumor cells [89], CSCs possess protective mechanisms against ROS-induced cytotoxicity [90]. This resilience contributes to their survival following treatment and highlights the potential of targeting ROS-dependent or redox-regulated pathways in CSCs to overcome therapy resistance and improve clinical outcomes [91].

ROS are primarily generated in the mitochondria through the electron transport chain (ETC) as natural byproducts of aerobic metabolism [92]. However, cancer cells exhibit additional sources of ROS, including mitochondrial dysfunction, increased enzymatic activity, oncogenic signaling, and the tumor microenvironment [93]. Mitochondrial dysfunction, particularly alterations in the ETC, leads to increased ROS generation. Upregulation or dysregulation of enzymes such as NADPH oxidase, xanthine oxidase, and cytochrome P450 also contributes significantly to elevated ROS levels in cancer cells [94]. Activation of oncogenes like KRAS and MYC enhances ROS production, often associated with cell proliferation and metabolic reprogramming [95]. The tumor microenvironment, characterized by conditions such as hypoxia and inflammation, further promotes ROS generation and modulates cellular metabolism and signaling pathways, influencing the overall redox state of cancer cells [96].

While elevated ROS levels can drive tumor progression and cellular signaling, excessive ROS can cause irreversible damage to DNA, proteins, and lipids, leading to apoptosis [97]. Consequently, cancer cells, including CSCs, must maintain a delicate balance between survival and proliferation while avoiding oxidative damage [98]. This balance presents a potential therapeutic target for developing more effective cancer treatments.

CSCs employ robust antioxidant systems to maintain redox homeostasis and counteract oxidative stress [99]. These systems include GSH, SOD enzymes, thioredoxin, and peroxiredoxin systems, and NADPH production through pathways like the pentose phosphate pathway (PPP) [100]. CSCs exhibit metabolic flexibility, adapting to oxidative stress by modulating their redox state. For instance, breast cancer stem cells show lower ROS levels due to upregulated antioxidant enzymes, contributing to their resistance and therapy evasion [101].

ROS play a dual role in CSC survival and tumor progression [102]. At moderate levels, they function as signaling molecules promoting cell proliferation, differentiation, and metastasis by activating pathways such as MAPK, PI3K, ERK, and NF-κB [102]. However, excessive ROS levels can lead to OS, DNA damage, and apoptosis.

To survive under OS, CSCs employ multiple mechanisms:Metabolic reprogramming (Warburg effect);Autophagy;Quiescence.

These adaptive strategies enable CSCs to maintain survival and contribute to tumor progression despite potentially harmful ROS levels [103]. The nuclear factor erythroid 2-related factor 2 (NRF2) pathway plays a crucial role in ROS signaling and cancer stem cell biology [39]. NRF2, a transcription factor governing antioxidant gene expression, protects CSCs from OS [104]. Its abnormal activation in various cancers contributes to therapy resistance and tumor recurrence, making the NRF2 pathway a promising therapeutic target for disrupting redox homeostasis in CSCs [105]. Recent research explored the role of ZMYND8, a histone reader, in regulating oxidative stress and ferroptosis in breast cancer stem cells (BCSCs) [106]. ZMYND8 is upregulated in BCSCs, reducing ROS and iron levels, thereby inhibiting ferroptosis [107]. It regulates NRF2 by increasing its protein stability and recruiting it to antioxidant gene promoters. A positive feedback loop between ZMYND8 and NRF2 amplifies antioxidant defense mechanisms, supporting BCSC survival and stemness [108]. In GSCs, Nrf2 expression is upregulated compared to non-stem glioma cells, leading to increased glutathione peroxidase 1 (GPx1) transcription and decreased ROS levels, which contributes to their radio-resistance [109]. Interestingly, microRNA-153 (miR-153) has been found to target Nrf2 in GSCs. Overexpression of miR-153 results in increased ROS production, radio-sensitization, and decreased stemness of GSCs [110]. This effect is mediated through the ROS-activated p38 MAPK pathway, which induces differentiation and reduces the neurosphere formation capacity of GSCs [110]. These findings highlight the complex interplay between Nrf2, ROS signaling, and cancer stem cell properties in glioblastoma.

NRF2 exhibits a context-dependent role in cancer, inhibiting tumor initiation in normal cells but promoting tumor progression in established tumors [109]. In BCSCs, ZMYND8-mediated NRF2 activation triggers a defense mechanism against oxidative stress and ferroptosis, supporting BCSC survival and stemness [106]. Hyperactivation of NRF2 is associated with various types of therapy resistance in cancer, including chemo- and radio-resistances, targeted therapy resistance, and immunotherapy resistance [111].

Controlled ROS levels are critical for CSC differentiation into endothelial cells (ECs) and tumor angiogenesis [101]. ROS-mediated signaling pathways, including hypoxia-inducible factor 1-alpha (HIF-1α) and vascular endothelial growth factor (VEGF), promote new blood vessel formation essential for tumor growth and metastasis [112]. Autophagy also plays a pivotal role in CSC differentiation into ECs, with ROS functioning as an initiator [110].

Many cancer therapies, including radiation and inhibitors of AKT, PI3K, and mTOR pathways, induce autophagy through ROS-mediated effects [113]. Elevated ROS levels drive metabolic shifts from glycolysis to the pentose phosphate pathway (PPP), enhancing nicotinamide adenine dinucleotide phosphate (NADPH) production and autophagy, further promoting CSC differentiation and angiogenesis [114].

CSCs typically maintain lower ROS levels compared to non-CSCs, supported by enhanced antioxidant defenses [115]. The NRF2 pathway plays a crucial role in this antioxidant capacity, contributing to therapy resistance and poor prognosis [115]. CSCs’ high antioxidant capacity confers resistance to ROS-inducing treatments like chemotherapy and radiotherapy [116].

Further research is needed to explore whether ZMYND8-mediated NRF2 activation in BCSCs can promote resistance to therapies in breast cancer [117]. Additionally, while the activation of NRF2 is known to play a crucial role in CSC resistance to OS, the downstream targets and pathways mediating this resistance are not fully mapped [117]. Addressing this knowledge gap could unveil new therapeutic opportunities.

Targeting the redox balance in cancer stem cells (CSCs) has emerged as a promising therapeutic strategy. The study of oxidative cell death mechanisms in cancer therapy has revealed innovative treatment possibilities [118]. Modifying the delicate balance of reactive oxygen species (ROS) in cancer cells can trigger selective cell death through various pathways, including ferroptosis, apoptosis, necroptosis, pyroptosis, parthanatos, oxeiptosis, and paraptosis [119,120]. Ferroptosis, a significant oxidative cell death mechanism, primarily relies on increasing ROS levels [121]. In ovarian cancer, for instance, FXN gene depletion elevates ROS levels, driving ferroptosis. The investigation of oxidative cell death holds promise for developing pharmacological agents to prevent tumorigenesis or treat established cancer. Targeting key antioxidant proteins, such as SLC7A11, GCLC, GPX4, TXN, and TXNRD, represents an emerging approach for inducing oxidative cell death in cancer cells [122].

CSCs employ various redox regulatory mechanisms to survive cancer treatments, contributing to therapeutic resistance and tumor recurrence. These mechanisms include enhanced DNA repair, metabolic reprogramming, microRNA-mediated redox regulation, and increased expression of drug efflux transporters [123,124]. To address this challenge, several strategies have been proposed to selectively disrupt CSC redox homeostasis:Inhibition of antioxidant systems: Drugs targeting glutathione (GSH) synthesis, superoxide dismutase (SOD) activity, or thioredoxin pathways can selectively impair CSC survival [84].ROS-inducing agents: Pro-oxidant therapies that elevate ROS levels beyond the tolerance threshold of CSCs can induce oxidative stress and apoptosis [125].Metabolic modulators: Agents that disrupt metabolic pathways involved in NADPH production, such as pentose phosphate pathway (PPP) inhibitors, can compromise CSC antioxidant defenses [126].

One promising compound is Resveratrol, which exerts antitumoral effects by inducing oxidative stress in breast cancer CSCs. It impairs mammosphere formation and tumor growth through autophagy induction and Wnt pathway inhibition [127]. In colorectal cancer, it enhances ROS production by overloading the mitochondrial electron transport chain, leading to increased apoptosis and cell death. Studies have demonstrated that resveratrol can enhance the antitumor effects of temozolomide (TMZ), a standard treatment for glioblastoma, by increasing ROS production [128]. This ROS-dependent mechanism activates the AMPK-TSC-mTOR signaling pathway, leading to cell cycle arrest, inhibition of cell migration, and downregulation of anti-apoptotic proteins [128].

Preclinical studies have demonstrated the potential of these strategies. Buthionine sulfoximine (BSO), a GSH synthesis inhibitor, sensitizes CSCs to chemotherapy by increasing ROS levels [129]. Similarly, arsenic trioxide (ATO) has shown efficacy in targeting leukemia stem cells through ROS-mediated mechanisms [130]. CSCs possess enhanced abilities to withstand redox stress and rapidly repair damaged DNA, which are crucial for their survival during cancer treatments [131]. They have amplified checkpoint activation and DNA damage repair mechanisms, allowing them to better cope with the oxidative stress induced by chemotherapy and radiation [132]. Additionally, CSCs demonstrate metabolic reprogramming, which is essential for maintaining their self-renewal potential and adapting to changes in the tumor microenvironment, including oxidative stress conditions [124].

MicroRNAs have been found to play a critical role in regulating therapeutic resistance through the modulation of antioxidant enzymes and redox-sensitive signaling pathways [133]. This microRNA-mediated redox regulation contributes to the CSCs’ ability to survive chemo/radiotherapy. Furthermore, CSCs exhibit increased expression of ATP-binding cassette transporters, which help expel anticancer drugs, further enhancing their resistance to chemotherapy [123].

Understanding these mechanisms is crucial for developing targeted therapies that can overcome CSC-mediated resistance and improve cancer treatment outcomes. However, several challenges remain in targeting redox balance in CSCs:Limited understanding of CSC antioxidant responses: The precise mechanisms through which CSCs modulate their antioxidant responses under different microenvironmental conditions remain poorly understood, hindering the development of targeted therapies.Therapeutic specificity: Developing agents that selectively target CSCs without affecting normal stem cells is critical to minimizing side effects.CSC heterogeneity: The diverse phenotypes and metabolic states of CSCs across different tumor types complicate therapeutic targeting.Resistance development: CSCs may develop resistance to redox-targeted therapies through compensatory pathways.

The complex interplay between OS and cancer stem cells (CSCs) presents both challenges and opportunities in cancer treatment. ROS play a dual role in cancer progression and cell death, while CSCs’ ability to maintain redox balance poses a significant obstacle to conventional therapies. The critical challenge lies in developing therapies that specifically target the redox balance in CSCs without affecting the redox homeostasis of normal stem cells.

## 4. Oxidative Stress in Neurodegenerative Diseases

NDDs are characterized by progressive loss of specific populations of neurons, leading to a decline in neurological function [134]. These disorders, including Alzheimer’s disease (AD), Parkinson’s disease (PD), Huntington’s disease (HD), and amyotrophic lateral sclerosis (ALS), pose an increasing threat to human health worldwide [134,135]. They are often associated with the formation of insoluble intracellular proteinaceous inclusions, suggesting a potential role of protein misfolding and aggregation in their pathogenesis [135]. Alzheimer’s disease is characterized by cognitive impairment and memory loss, with amyloid-β deposition being a key pathological feature [136]. Parkinson’s disease primarily affects movement, resulting in tremors, rigidity, and bradykinesia [137]. Huntington’s disease is an inherited disorder causing progressive brain damage and is associated with motor, cognitive, and psychiatric symptoms [135]. Amyotrophic lateral sclerosis, also known as motor neuron disease, leads to the degeneration of motor neurons, resulting in muscle weakness and atrophy. While the exact mechanisms underlying these diseases remain unclear, research has implicated various factors such as apoptosis, oxidative stress, and mitochondrial dysfunction in their pathogenesis [137]. The aging brain is a major risk factor for most neurodegenerative disorders, and with the growing elderly population, these diseases have become a significant health concern [137]. Despite extensive research efforts, effective disease-modifying therapies remain elusive, highlighting the need for continued investigation into the molecular pathways and potential therapeutic targets for these devastating conditions.

OS plays a crucial role in the pathogenesis of NDDs [138]. The brain is particularly vulnerable to oxidative damage due to its high oxygen demand and abundance of peroxidizable substrates [139]. In NDDs such as Alzheimer’s, Parkinson’s, Huntington’s, and amyotrophic lateral sclerosis, increased levels of oxidative stress have been observed [140]. This OS can lead to damage of nucleic acids, proteins, and lipids, as well as mitochondrial dysfunction, glial cell activation, and inflammatory responses. Interestingly, while oxidative stress is strongly linked to neurodegeneration, it remains unclear whether it is a primary cause or a secondary manifestation of the disease process [141]. The main sources of OS in NDDs include mitochondrial dysfunction, which is a major endogenous source of ROS [142]. Additionally, the phagocyte NADPH oxidase (PHOX) and non-phagocyte NADPH oxidases in astroglia and neurons have been identified as significant contributors to oxidative stress in NDs [143]. The cerebrovascular architecture has also been implicated as a potential source of oxidative stress, highlighting the importance of vascular-derived oxidative stress in brain disorders [144,145]. Understanding these sources and mechanisms of oxidative stress is crucial for developing effective therapeutic strategies for neurodegenerative diseases.

OS in NDDs arises from multiple interconnected sources, with mitochondrial dysfunction playing a central role. The electron transport chain (ETC), particularly complexes I and III, generates ROS that damage cellular components. This damage initiates a vicious cycle of increasing OS, impaired ATP production, and altered calcium handling [142]. The presence of iron in mitochondria exacerbates this damage through hydroxyl radical formation [146].

Neuroinflammation significantly contributes to OS generation. Chronic activation of microglia and astrocytes in neurodegenerative conditions leads to the release of pro-inflammatory mediators and ROS, creating a neurotoxic environment that promotes disease progression [147].

Excitotoxicity, resulting from excessive glutamate signaling, is another major source of OS [148]. Over-activation of glutamate receptors triggers a cascade of events, including calcium dysregulation, mitochondrial dysfunction, and ROS generation. The bidirectional relationship between excitotoxicity and OS creates a self-perpetuating cycle of neuronal damage [149].

Iron dysregulation and ferroptosis have emerged as significant contributors to OS in NDDs, particularly in Alzheimer’s disease (AD) and Parkinson’s disease (PD) [150]. Iron accumulation in affected brain regions has been implicated in pathogenesis, leading to neuronal dysfunction and cell death [151]. While AD and PD have distinct etiopathogeneses, they share common cellular processes related to excessive iron deposition, iron-induced oxidative stress, and the accumulation of lipid-generated ROS.

In both AD and PD, iron overload contributes to ROS generation and lipid peroxidation, which are key features of ferroptosis [152]. In AD, iron accumulation is associated with Aβ-plaque formation and phosphorylated tau, while in PD, it is linked to progressive dopaminergic neuronal loss [153]. Ferroptosis, an iron-dependent form of regulated cell death, plays a crucial role in the pathogenesis of neurodegenerative diseases [154]. The interplay between iron dysregulation, lipid peroxidation, and mitochondrial dysfunction contributes to disease progression. Understanding these mechanisms provides new insights into potential therapeutic targets, with anti-ferroptotic agents and iron chelators showing promise in preclinical and clinical studies for treating NDDs [155]. The interplay between these mechanisms—mitochondrial dysfunction, neuroinflammation, excitotoxicity, ferroptosis, and iron dysregulation—creates a complex network of pathological processes that contributes to neurodegeneration.

### Oxidative Stress-Driven Mechanisms in NDDs

Oxidative stress and mitochondrial dysfunction are intricately linked in the pathogenesis of NDDs [156]. Mitochondria, the primary source of reactive oxygen species (ROS), play a crucial role in initiating and perpetuating oxidative stress in neurons. ROS overproduction can lead to mitochondrial DNA mutations, respiratory chain damage, altered membrane permeability, and disrupted Ca^2+^ homeostasis, contributing to neuronal dysfunction and eventual neurodegeneration [157].

A feedback loop between oxidative stress and mitochondrial dysfunction has been proposed in Alzheimer’s disease. Cyclin-dependent kinase 5 (Cdk5) dysregulation can lead to ROS accumulation, causing mitochondrial damage, which further increases ROS production, creating a vicious cycle resulting in cell death [158]. Despite promising results in experimental models, clinical trials targeting general oxidative stress have not yet demonstrated a significant impact on disease progression, which highlights the need for more targeted approaches [159].

Oxidative stress contributes to neurodegeneration through various mechanisms, including damage to neuronal membranes rich in polyunsaturated fatty acids (PUFAs) and the formation of toxic by-products [160]. The central nervous system (CNS) is particularly vulnerable due to its high oxygen consumption, elevated iron content in certain brain areas, and abundance of PUFAs in neuronal membranes. ROS can initiate lipid peroxidation (LPO), targeting PUFAs and producing toxic lipid aldehyde species such as 4-hydroxy-2-nonenal (HNE), malondialdehyde, and acrolein [161]. These reactive aldehydes cause post-transcriptional modifications of DNA and proteins, resulting in genotoxicity, inhibition of gene expression, cytotoxicity, and cellular death.

Interestingly, while PUFAs are susceptible to oxidative damage, they also play a protective role in neuronal health. The omega-3 and omega-6 fatty acids protect neurons from endoplasmic reticulum (ER) and oxidative stress, suppressing apoptosis in certain neurodegenerative conditions [162]. Dietary supplementation with omega-3 PUFAs has demonstrated therapeutic potential in neurotrauma, reducing neuroinflammation and oxidative stress while promoting cell survival pathways [163].

Oxidative stress plays a significant role in promoting the aggregation of misfolded proteins, such as amyloid-β, tau, and α-synuclein, which are key contributors to various neurodegenerative disorders. These misfolded proteins, in turn, induce further oxidative damage, creating a vicious cycle of neurodegeneration [164]. In Alzheimer’s disease, amyloid-β and tau proteins interact with mitochondria, leading to dysfunction and increased production of reactive oxygen species (ROS) [165]. Similarly, in Parkinson’s disease, α-synuclein aggregation is linked to oxidative stress and inflammation, reducing brain mitochondrial activity [166].

The interplay between oxidative stress and protein aggregation is complex and multifaceted. While oxidative stress promotes protein aggregation, the aggregated proteins themselves can induce further oxidative damage [167]. For instance, both oligomeric and fibrillar forms of α-synuclein generate free radicals, but only the oligomeric form leads to glutathione reduction and subsequent neuronal toxicity [168]. Additionally, free metal ions play a crucial role in oligomer-induced ROS production, as metal chelators can block this process and prevent neuronal death [169].

ROS significantly contribute to the production and aggregation of amyloid beta (Aβ) in Alzheimer’s disease [170]. Redox-active metal ions catalyze ROS production when bound to Aβ, contributing to oxidative damage of the peptide itself and surrounding molecules. This OS increases Aβ production and aggregation, promoting AD progression [170]. Interestingly, the literature suggests that Aβ plaques and neurofibrillary tangles may function as a primary line of antioxidant defense, which occurs because of OS rather than initiating disease pathogenesis [165]. This contradicts the traditional view of these structures as central mediators of AD pathogenesis and raises concerns about current therapeutic approaches focused on lesion removal.

OS also contributes to tau hyperphosphorylation and neurofibrillary tangle formation. The exacerbated production of ROS triggers oxidative stress, leading to tau hyperphosphorylation [171]. This process can result in neurofibrillary tangle formation, inducing further production of cytotoxic ROS and causing neuronal apoptosis. The relationship between oxidative stress, tau hyperphosphorylation, and autophagy dysregulation in AD remains complex and is not fully understood. In conclusion, oxidative stress-driven mechanisms in neurodegeneration involve a complex interplay between ROS, PUFAs, and their oxidation products and misfolded proteins [172]. Understanding these mechanisms is crucial for developing targeted therapies and identifying potential biomarkers, such as isoprostanes and neuroprostanes, for NDDs.

Targeting oxidative stress pathways has emerged as a promising approach to mitigate neurodegeneration, with two key strategies showing potential: activation of Nrf2 and inhibition of NADPH oxidases [173,174]. Nrf2 activation enhances endogenous antioxidant systems, playing a crucial role in counteracting oxidative stress and neuroinflammation in neurodegenerative disorders [175]. Several studies have demonstrated the neuroprotective effects of Nrf2 activation:Kavalactones attenuate amyloid beta-peptide toxicity by inducing Nrf2-mediated protective gene expression in vitro [176].Tert-butylhydroquinone treatment and adenoviral Nrf2 gene transfer protect against amyloid beta toxicity in Alzheimer’s disease models [177].Food-derived Nrf2/ARE pathway inducers, such as l-sulforaphane from broccoli and isoliquiritigenin from licorice, protect mitochondrial function in oxidative stress and neurodegenerative disease models [174]. The Nrf2-ARE pathway not only addresses OS but also modulates mitochondrial function, reduces neuroinflammation, and promotes neuroprotection.

NADPH oxidase inhibitors offer another promising approach for mitigating neurodegeneration. NADPH oxidases are enzymes dedicated to ROS production, playing a key role in oxidative stress-related neurological disorders. Studies have shown the following:Apocynin, an NADPH oxidase inhibitor, attenuates microglial activation, oxidative stress damage, and induction of Alzheimer’s disease proteins in traumatic brain injury models [178].Inhibiting NADPH oxidases may reduce ROS-mediated damage to retinal ganglion cells and glial dysfunction in glaucoma [179].

However, challenges remain in developing isoform-selective NADPH oxidase inhibitors. Current inhibitors like GKT137831, ML171, and VAS2870 show improved specificity but only moderate isoform selectivity [180]. Additionally, the role of ROS in brain repair is not fully understood, potentially impacting the long-term effects of NADPH oxidase inhibition. In conclusion, both Nrf2 activation and NADPH oxidase inhibition present promising strategies for mitigating neurodegeneration by targeting oxidative stress pathways. As research progresses, the development of specific Nrf2 activators, enhancers of endogenous antioxidant systems, and more selective NADPH oxidase inhibitors may provide novel therapeutic interventions for a range of NDDs.

## 5. Common Mechanisms in CSCs and Neurodegenerative Diseases

Aging is characterized by a progressive accumulation of cellular damage, impaired physiological functions, and increased vulnerability to diseases such as cancer and neurodegeneration [181]. Despite their distinct characteristics (Table 2), uncontrolled proliferation in cancer versus progressive cellular loss in neurodegeneration share mechanisms rooted in the hallmarks of aging, such as oxidative stress, genomic instability, telomere attrition, epigenetic changes, proteostasis loss, mitochondrial dysfunction, and altered intercellular communication (Figure 2) [182,183].

OS and redox imbalance are critical factors that drive the pathogenesis of both cancer and NDDs. In neurodegenerative diseases, elevated reactive oxygen and nitrogen species (ROS/RNS) damage cellular components and drive diseases such as Alzheimer’s and Parkinson’s disease [197,198]. Conversely, in cancer, OS promotes DNA mutations and pro-oncogenic signaling, aiding tumor initiation and progression [97] (Figure 3). For instance, in leukemia, high ROS production is associated with decreased antioxidant defense, leading to cellular damage [199,200]. Chronic oxidative stress has also been implicated in tumor progression and drug resistance. Interestingly, oxidative stress has dual roles; while it can cause cellular damage, it can also be protective under certain conditions. For example, NOX4, a NADPH oxidase known to produce ROS, has been found to be neuroprotective by regulating ROS and calcium homeostasis, thereby preventing hyperexcitability and neuronal death [201]. This suggests that some ROS-producing enzymes might serve as redox regulators, potentially benefiting neurons in hyperexcitable states associated with neurodegeneration. In chemotherapy, elevated reactive oxygen species (ROS) levels can induce oxidative stress-mediated apoptosis in cancer cells, potentially inhibiting tumor growth [202]. A few chemotherapeutic drugs induce oxidative stress and exert biological activity against cancer cells.

CSCs exploit oxidative stress by enhancing their antioxidant defense mechanisms, allowing them to thrive in adverse conditions. In contrast, cells affected by neurodegenerative diseases often suffer from impaired antioxidant responses, leading to cell damage and disease progression. Understanding these divergent roles offers therapeutic potential: antioxidants and targeted pathways such as HO-1/PARP1 for neurodegeneration and oxidative stress modulation to enhance cancer treatments. These insights underscore the shared yet distinct impact of aging mechanisms on cancer and neurodegenerative diseases.

### 5.1. Oxidative Stress: Overlap Between Mechanisms in Cancer Stem Cells and Neurodegeneration

#### 5.1.1. Oxidative Stress and Mitochondrial Dysfunction

Oxidative stress and mitochondrial dysfunction are closely interrelated and exacerbate each other in various pathological conditions [140]. Cancer stem cells (CSCs) exhibit distinct metabolic reprogramming, including increased mitochondrial biogenesis, enhanced oxidative phosphorylation (OXPHOS), and low reactive oxygen species (ROS) levels. These adaptations, achieved through FoxM1-dependent Prx3 expression and fatty acid oxidation-mediated NADPH regeneration, support CSC stemness, self-renewal, and survival [203]. The hypoxic tumor microenvironment further promotes stemness pathways and ROS regulation in CSCs, highlighting the role of metabolic reprogramming in tumor progression and resistance to treatment [112]. In conclusion, metabolic reprogramming observed in CSCs, characterized by increased mitochondrial biogenesis and altered ROS regulation, is critical for maintaining stemness, self-renewal, and antioxidant defense mechanisms [204]. These metabolic features of CSCs are crucial for the development of targeted therapies to overcome tumor progression and resistance to treatment.

In contrast, neurodegenerative diseases are marked by early mitochondrial dysfunction and oxidative stress, which impair energy production and synaptic transmission, particularly in neurons owing to their high metabolic activity [205]. For example, Alzheimer’s disease (AD) is characterized by reduced complex IV activity and increased oxidative damage, while Huntington’s disease (HD) exhibits reductions in complex II activity, increased cortical lactate levels, and oxidative damage. In amyotrophic lateral sclerosis (ALS), familial cases are linked to mutations in the Cu, Zn superoxide dismutase (SOD1) gene, whereas sporadic cases demonstrate increased oxidative damage [206]. Mitochondrial defects and increased ROS generation create a feedback loop, driving neuronal loss and dysfunction [207]. The interplay highlights the need for therapeutic strategies targeting mitochondrial function and ROS regulation to preserve neuronal health.

An intricate balance between oxidative stress and mitochondrial function underscores the potential for targeting metabolic pathways to address cancer progression and neurodegeneration. Further, we explored how redox signaling influences cellular survival, highlighting the dual roles of key transcription factors in both CSCs and neurodegeneration, Table 3.

#### 5.1.2. Redox Signaling and Cellular Survival

Redox signaling is pivotal for cellular survival, influencing cancer stem cells (CSCs) and neurodegenerative diseases [223]. Transcription factors like nuclear factor erythroid 2-related factor 2 (Nrf2) and nuclear factor kappa-light-chain-enhancer of activated B cells (NF-κB) regulate oxidative stress and inflammation, albeit in a context-dependent manner [224,225].

Nrf2 activation in CSCs promotes tumorigenicity, chemoresistance, and cytoprotection, while its attenuation in neurodegenerative diseases like AD increases OS and neuroinflammation [226]. Nrf2-targeted therapies, such as RNAi in lung cancer and rifampicin in neurodegeneration, exemplify its dual role and therapeutic potential [227]. Similarly, NF-κB signaling sustains CSC survival and proliferation, while its inhibition reduces neuroinflammation in neurodegenerative diseases. Compounds like bis(ethylmaltolato)oxidovanadium (BEOV) and evodiamine (EV) suppress NF-κB activation, which demonstrates their neuroprotective effects [228,229].

Thiol-redox proteins such as glutaredoxin (Grx) and thioredoxin (Trx) are integral to maintaining redox homeostasis in both CSCs and neurodegeneration [89]. Trx protects neurons against oxidative stress, with decreased levels observed in Parkinson’s disease (PD) and AD [230]. In cancer, these systems enable CSCs to resist redox stress, highlighting their therapeutic relevance. These findings underscore the intricate role of redox signaling in cellular survival, emphasizing the need for targeted, context-specific therapeutic interventions to address both cancer progression and neurodegenerative diseases.

#### 5.1.3. Oxidative Stress and Chronic Inflammation

Oxidative stress plays a pivotal role in both cancer stem cells (CSCs) and neurodegenerative diseases, driving inflammation and pathological outcomes. In neurodegenerative diseases, oxidative stress contributes to neuroinflammation and neuronal death by damaging lipids, proteins, and nucleic acids, which results in mitochondrial dysfunction and activation of inflammatory pathways [231]. This damage triggers excitotoxicity and calcium overload, culminating in neuronal cell death [232]. A positive feedback loop between oxidative stress and neuroinflammation accelerates neuron loss in conditions like Alzheimer’s and Parkinson’s diseases [233]. Reactive oxygen and nitrogen species (ROS and RNS) exacerbate this cycle, perpetuating neuronal damage and inflammation. The TNF pathway exemplifies this synergy, as TNF propagates inflammation via TNFR1 while simultaneously inducing oxidative stress by activating ROS- and RNS-producing enzymes [234].

In CSCs, the interplay between ROS and inflammatory pathways is multifaceted. Excessive ROS activates transcription factors such as NF-κB, p53, HIF-1α, and AP-1, leading to the expression of pro-inflammatory cytokines and growth factors that support cancer cell survival and proliferation [235]. While CSCs generally resist oxidative stress better than non-CSCs, high ROS levels can induce senescence, as observed in breast cancer stem cells exposed to sublethal H_2_O_2_ doses, which activate the p53/p21 signaling pathway [129].

Matrix metalloproteinases (MMPs) further link oxidative stress to neurodegeneration and cancer [236]. In multiple sclerosis, elevated MMP activity damages the blood–brain barrier (BBB) and myelin. Similarly, in cancer, MMPs regulate tumor-specific gene activation. Oxidative stress activates MMPs (e.g., MMP-1, -2, -9) and reduces their inhibitors (TIMPs), leading to BBB disruption and increased permeability [237,238]. Interestingly, MMPs can act on intracellular substrates rapidly, suggesting novel roles in oxidative stress responses [239].

Therapeutic approaches targeting oxidative stress and MMP activity are promising but challenging. Antioxidants like superoxide dismutase (SOD) have shown efficacy in experimental models, though clinical trials yield mixed results, necessitating targeted interventions [240]. MMP inhibition offers potential for mitigating oxidative stress-induced damage, particularly in cardiac and neural tissues. Future strategies must balance free radical scavenging while maintaining essential ROS-dependent functions, considering genetic and environmental factors. In summary, the intricate interplay between oxidative stress, inflammation, and MMPs underscores their central roles in neurodegeneration and cancer. Multifaceted therapeutic approaches targeting these interconnected pathways hold promise for addressing these chronic conditions effectively.

#### 5.1.4. Oxidative Stress: Lipid Peroxidation and Ferroptosis

Accumulation of lipid peroxides due to oxidative stress plays a crucial role in ferroptosis, a form of regulated cell death characterized by iron-dependent lipid peroxidation [241]. In neurodegenerative diseases, this process contributes to neuronal death, while cancer stem cells (CSCs) employ protective mechanisms against ferroptosis to maintain survival. In neurodegenerative diseases, ferroptosis is emerging as a key player in neuronal cell death. For instance, in Parkinson’s disease (PD), the accumulation of lipid peroxides, iron deposition, and enhanced oxidative stress contribute to the pathogenic mechanism [242]. Similarly, in Friedreich’s Ataxia (FA), iron-induced oxidative damage accumulates over time, lowering the ferroptosis threshold and leading to neuronal cell death [243]. Ferroptosis has also been implicated in other neurological disorders such as intracerebral hemorrhage, ischemic stroke, and epilepsy [244]. Interestingly, while ferroptosis contributes to neurodegeneration, CSCs exhibit protection against this process to maintain their survival [245]. CSCs often have very active lipid and iron metabolism, which paradoxically makes them more susceptible to ferroptosis. However, they have developed mechanisms to resist ferroptosis, making it a potential target for cancer therapy [110]. Ferroptosis inducers could specifically induce CSC death in tumors, potentially overcoming cancer resistance [246,247]. This understanding opens up new therapeutic avenues, such as anti-ferroptosis drugs for neurodegenerative diseases and ferroptosis inducers for targeting CSCs in cancer treatment.

##### Iron Metabolism and Lipid Peroxidation: Contrasting Roles in CSCs and Neurons

Iron metabolism and lipid peroxidation play crucial roles in both cancer stem cells (CSCs) and neurons but with distinct differences in their regulation and effects. In CSCs, iron metabolism is signi ficantly altered to support their high proliferation and survival rates. CSCs exhibit increased iron uptake and storage compared to non-CSCs, which contributes to their enhanced growth and self-renewal capabilities. This elevated iron content makes CSCs more susceptible to ferroptosis, a form of cell death caused by iron-dependent lipid peroxidation [248]. The higher iron levels in CSCs also lead to increased production of reactive oxygen species and oxidative stress, which can promote tumor growth and metastasis [249]. In contrast, neurons are particularly vulnerable to iron-induced oxidative damage. Excessive iron accumulation in the brain is associated with neuronal damage and has been linked to the severity of stroke. Unlike CSCs, neurons do not benefit from increased iron levels and instead require tight regulation of iron homeostasis to prevent oxidative stress and cell death [250]. Lipid peroxidation in CSCs is closely tied to their unique metabolic characteristics. CSCs show alterations in lipid metabolism, including increased fatty acid uptake, de novo lipogenesis, and formation of lipid droplets [251]. These changes in lipid metabolism make CSCs more susceptible to ferroptosis when iron levels are elevated. In summary, while both CSCs and neurons are affected by iron metabolism and lipid peroxidation, CSCs have adapted to utilize increased iron levels for their survival and growth, whereas neurons are more sensitive to iron-induced oxidative damage. Understanding these differences could lead to targeted therapies for both cancer and neurological disorders.

##### Polyunsaturated Fatty Acids (PUFAs) and Ferroptosis Susceptibility

Lipid peroxidation, particularly of polyunsaturated fatty acids (PUFAs), plays a crucial role in ferroptosis, a form of regulated cell death characterized by iron-dependent accumulation of lipid peroxides [252]. PUFAs are highly susceptible to oxidation, and their incorporation into cellular membranes represents a vulnerability to oxidative damage [252]. In cancer stem cells (CSCs) and neurons, PUFAs modulate oxidative damage through various mechanisms. Interestingly, the relationship between PUFA levels and ferroptosis susceptibility is not straightforward. While extracellular lipid limitation reduces cellular PUFA levels, it paradoxically increases ferroptosis sensitivity in cancer cells. This is due to a fatty acid trafficking pathway that liberates PUFAs from triglycerides to synthesize highly unsaturated PUFAs like arachidonic and adrenic acid, which accumulate in phospholipids and promote ferroptosis sensitivity [253]. In neurons, the depletion of diacyl-PUFA phosphatidylcholines (PC-PUFA2s) has been observed in aging and Huntington’s disease brain tissue, linking it to ferroptosis. PC-PUFA2s interact with the mitochondrial electron transport chain, generating reactive oxygen species (ROS) that initiate lipid peroxidation [254]. This suggests a critical role for PC-PUFA2s in controlling mitochondrial homeostasis and ferroptosis in various contexts, including neuronal cells. In summary, PUFAs modulate oxidative damage in CSCs and neurons through their incorporation into membrane phospholipids, their trafficking between different lipid pools, and their interaction with cellular organelles like mitochondria. The balance between different types of fatty acids (e.g., PUFAs vs. monounsaturated fatty acids) in membrane phospholipids is crucial in determining cellular sensitivity to ferroptosis. Understanding these mechanisms can provide insights into potential therapeutic strategies for ferroptosis-related diseases in both cancer and neurological contexts.

#### 5.1.5. DNA Damage and Genomic Instability

ROS-induced DNA damage plays a significant role in both cancer stem cells (CSCs) and neurodegenerative diseases, albeit with different outcomes [255]. In CSCs, ROS-induced DNA damage drives mutations that enhance self-renewal and therapy resistance. CSCs possess amplified checkpoint activation and DNA damage repair mechanisms, which allow them to survive and initiate tumor recurrence following chemotherapy and radiation treatments. For instance, in chronic myeloid leukemia (CML) stem cells, Rac2 GTPase alters mitochondrial function, leading to high ROS production through the mitochondrial respiratory chain complex III (MRC-cIII) [256]. This MRC-cIII-generated ROS promotes oxidative DNA damage, triggering genomic instability and accumulation of chromosomal aberrations. Similarly, BCR-ABL1 oncogene in CML cells induces increased ROS production, which can lead to DNA lesions and genomic instability [257]. Additionally, CSCs have increased abilities to detoxify or mediate the efflux of cytotoxic agents and resist oxidative stress, further contributing to their therapeutic resistance. Interestingly, while elevated ROS levels can induce chromosomal aberrations and mitochondrial DNA damage in stem cells, physiological levels of ROS are actually required to maintain their self-renewal capacity [124]. This highlights the complex relationship between ROS and stem cell function.

In contrast, for neurodegenerative diseases, ROS-induced DNA damage impairs repair mechanisms and leads to neuronal dysfunction [258]. In AD, increased oxidative DNA damage has been observed in both nuclear and mitochondrial DNA extracted from post-mortem brain regions [259]. Additionally, cytogenetic damage, such as a tendency toward chromosome 21 malsegregation, has been reported in AD patients [258]. Parkinson’s disease (PD) and amyotrophic lateral sclerosis (ALS) also exhibit elevated levels of oxidative DNA damage, which is considered one of the earliest detectable events in neurodegeneration [136]. Interestingly, the relationship between DNA damage and neurodegeneration is bidirectional. While oxidative stress can cause DNA damage, the accumulation of DNA damage can also lead to increased ROS production, creating a vicious cycle. Furthermore, recent studies have revealed an emerging interplay between environmental-induced oxidative stress and epigenetic modifications of critical genes for neurodegeneration, which adds another layer of complexity to the genomic instability observed in these disorders [260]. The low regeneration capacity of neurons and insufficient secretion of neurotrophic factors exacerbate the damage caused by oxidative stress in the central nervous system [261]. In addressing these issues, antioxidant therapies have shown promise in both fields. In cancer treatment, naturally occurring polyphenols with antioxidant properties have emerged as promising anticancer compounds, targeting epigenetic processes and inducing ROS-dependent premature senescence in cancer cells [262]. For neurodegenerative diseases, antioxidant supplementation has been found to mitigate oxidative stress, improve stem cell survival, and enhance neurogenesis in patients with stroke and neurodegenerative conditions [263].

#### 5.1.6. Aging-Related Oxidative Damage in CSCs and NDDs

Aging is closely linked to oxidative stress, creating a self-reinforcing cycle that contributes to cellular damage and dysfunction. This imbalance arises from increased reactive oxygen and nitrogen species (RONS) production coupled with diminished antioxidant defenses, leading to cumulative damage to DNA, lipids, and proteins [264]. The physiological aging process is inherently associated with elevated oxidative stress levels, which in turn accelerate aging by impairing cellular homeostasis [265]. The free radical theory of aging suggests that free radicals generated through oxygen-associated reactions drive the aging process [265]. This interplay between aging and oxidative stress underlies various age-related complications, including mitochondrial dysfunction, genomic instability, and chronic inflammation. Understanding these mechanisms is essential for developing strategies to mitigate age-associated diseases.

One major consequence of age-related oxidative stress is its impact on cancer stem cells (CSCs). Reactive oxygen species (ROS) play a pivotal role in stem cell self-renewal and differentiation during development and organogenesis [186]. However, aging disrupts redox homeostasis, leading to oxidative damage accumulation and cellular senescence, thereby impairing regenerative capacity in the cardiovascular system and other tissues [266]. Interestingly, CSCs maintain significantly lower ROS levels than bulk tumor cells, allowing them to resist chemotherapy and radiotherapy [224]. This resistance is mediated through the upregulation of nuclear factor erythroid 2-related factor 2 (Nrf2), a key regulator of antioxidant responses, which promotes CSC survival, tumor progression, and metastatic potential [224]. Age-induced oxidative stress may further enhance cancer progression by activating genetic reprogramming and stemness pathways in senescent cells, ultimately contributing to tumor recurrence and metastasis [264]. Additionally, prolonged oxidative damage fosters genomic instability, potentially generating new CSC populations with heightened malignancy.

Neurons, due to their high metabolic activity and limited oxidative damage repair capacity, are particularly susceptible to oxidative stress [267]. The brain’s substantial metabolic demands lead to progressive oxidative damage accumulation over time, exacerbated in cases of impaired DNA repair, which can contribute to neurological dysfunction [267]. Aging further intensifies oxidative damage in neurodegenerative conditions by disrupting the balance between ROS/RNS generation and detoxification [137]. While ROS/RNS play essential roles in cellular signaling at physiological levels, excessive accumulation triggers oxidative modification and functional impairment of proteins, nucleic acids, and lipids [268]. Age-related alterations in microRNA biogenesis further increase neuronal vulnerability, potentially accelerating neurodegenerative disease progression [268]. Notably, certain neuron populations exhibit greater susceptibility to oxidative stress due to intrinsic factors such as high oxidative load, mitochondrial dysfunction, and heightened inflammatory responses [268]. Additionally, aging impairs autophagy, reducing the clearance of damaged cellular components and promoting neurodegeneration [269]. Understanding these interconnected mechanisms is crucial for developing neuroprotective strategies to combat aging-related cognitive decline and neurodegenerative diseases.

#### 5.1.7. Epigenetic Consequences of Oxidative Stress

Epigenetic mechanisms play a crucial role in the complex interplay between oxidative stress and cancer development. A study on non-tumoral kidney cells demonstrated that continuous exposure to oxidative stress can lead to malignant transformation, highlighting the profound impact of redox imbalance on cellular fate [270]. This transformation process involves significant alterations in the expression of both histone-modifying enzymes (HDAC1, HMT1, and HAT1) and key epigenetic regulators (DNMT1, DNMT3a, and MBD4). Importantly, the acquired tumorigenic potential of these transformed cells was reduced following treatment with the DNA demethylating agent 5-aza-2′-deoxycytidine, underscoring the critical involvement of epigenetic machinery in tumor development [271].

A major link between oxidative stress and epigenetics is glutathione (GSH) metabolism, which influences DNA methylation and histone modifications [272]. GSH depletion leads to global DNA hypomethylation due to reduced S-adenosylmethionine (SAM) availability [271]. Since SAM serves as a key methyl donor for DNA methyltransferases (DNMTs) and histone methyltransferases (HMTs), disruptions in its synthesis alter epigenetic patterns [273]. The enzymes responsible for SAM synthesis, methionine adenosyltransferase (MAT) and methionine synthase (MS), are highly sensitive to oxidative stress. This sensitivity results in decreased methyltransferase activity and genomic hypomethylation under redox imbalance, further promoting tumor development [271].

Oxidative stress affects epigenetics through multiple mechanisms, including changes in DNA methylation, histone modifications, and non-coding RNA expression. Reactive oxygen species (ROS) can modify DNMT activity, leading to abnormal methylation patterns that silence tumor suppressor genes [274]. High ROS levels also disrupt chromatin structure by altering histone deacetylases (HDACs) and acetyltransferases (HATs), impacting gene expression. Additionally, oxidative stress regulates microRNA (miRNA) and long non-coding RNA (lncRNA) activity, influencing pathways such as NRF2–ARE detoxification, which plays a protective role against oxidative damage [275].

Recent studies have shown that oxidative stress directly affects epigenetic chromatin modifications that control how genes are expressed. These changes may drive physiological responses to oxidative stress and facilitate the progression of diseases, including neurodegeneration [276]. Oxidative stress can directly reduce the methylation of DNA by oxidizing DNA, increasing TET-mediated hydroxymethylation, and interfering with the binding of DNA methyltransferases that produce the methyl donor S-adenosylmethionine [277]. ROS can also form oxidized DNA lesions by hydroxylating pyrimidines and 5-methylcytosine (5mC), which can interfere with 5-hydroxymethylcytosine (5hmC) epigenetic signals [278]. Oxidative stress alters post-translational histone modifications, which can change chromatin structure, gene expression, gene stability, and replication. These effects are often indirect, as ROS impair metabolic efficiency, reducing levels of metabolites such as acetyl-CoA, Fe, NAD+, and ketoglutarate that are essential for histone-modifying enzymes [279]. H_2_O_2_, the most common means of ROS induction in vitro, has been shown to increase H3K9me3, H3K4me3, and H3K27me3 while decreasing H3K9ac and H4K8ac in bronchial epithelial cells [276]. Preincubation with ascorbate prevented this elevation, indicating that antioxidants can prevent ROS-induced epigenetic changes. The methylation effects of oxidation were transient and did not persist after the H_2_O_2_ washout [276]. In neuronal SH-SY5Y cells, H_2_O_2_ increased histone acetyltransferases and downregulated histone deacetylases, as well as hypomethylating APP and BACE1 [276]. H_2_O_2_ thus caused the upregulation of the APP and BACE1 gene promoters, leading to amyloid beta peptide overproduction. These findings suggest that oxidative stress generates transient epigenetic modifications, which, among its wide-reaching effects, can accelerate amyloid beta pathology.

Oxidative stress plays a key role in aging-related epigenetic changes, particularly in upregulating the repressive histone mark H3K9me3 [280]. Research has shown that H3K9me3 levels increase with age in the hippocampus, and inhibiting its primary enzyme, SUV39H1, improves cognitive function and synaptic density in aged mice [280]. The Vaquero group suggests that oxidative stress elevates H3K9me3 indirectly by upregulating SIRT1, which stabilizes SUV39H1 [280]. Additionally, inhibition of another histone methyltransferase, G9a, has shown neuroprotective effects in Alzheimer’s disease (AD) models, improving memory and reducing amyloid beta plaques [281]. These findings indicate that oxidative stress may drive H3K9 methylation, and inhibiting this modification could mitigate cognitive decline and AD pathology. However, further research is needed to clarify the specific mechanisms involved.

#### 5.1.8. Autophagy and Oxidative Stress

Autophagy plays a dual role in oxidative stress management, particularly in cancer stem cells (CSCs) and neurodegeneration. In CSCs, autophagy contributes to both stemness maintenance and loss [282]. Most researchers believe that autophagy helps maintain CSC stemness and is responsible for anticancer therapy failure. However, some studies suggest that autophagy can also mediate the loss of stemness in CSCs [283]. This dual role is further complicated by the relationship between autophagy and reactive oxygen species (ROS). High levels of ROS can increase autophagy by inhibiting glucose-6-phosphate dehydrogenase and inactivating the pentose phosphate pathway, potentially influencing the differentiation of CSCs into tumor endothelial cells [283]. In neurodegeneration, the impairment of autophagy plays a crucial role. Excessive or insufficient autophagic activity in neurons leads to altered homeostasis and influences their survival rate, causing neurodegeneration [284]. Autophagy is required for maintaining cellular homeostasis under multiple stresses, but overactivation or inhibition can disrupt the homeostatic degradation of proteins and organelles within the brain, contributing to neuronal cell death [285]. Interestingly, while autophagy impairment is generally associated with neurodegeneration, excessive autophagosome formation has been observed early during necrotic cell death in C. elegans, suggesting that autophagy can also contribute to cellular destruction during necrosis [286]. In conclusion, the dual role of autophagy in oxidative stress management highlights its complex nature in both CSCs and neurodegeneration. Understanding the fine balance of autophagy in these contexts is crucial for developing targeted therapies. Future research should be focussed on elucidating the molecular mechanisms linking oxidative stress and autophagy to better comprehend their roles in disease progression and potential treatment strategies.

#### 5.1.9. Oxidative Stress and Nrf2

Nuclear factor erythroid 2-related factor 2 (Nrf2) is a key transcription factor that plays a central role in cellular defense against oxidative and electrophilic stress [287]. By binding to the antioxidant response element (ARE) in promoter regions, Nrf2 regulates the expression of antioxidative and phase-2 detoxifying enzymes, as well as other cytoprotective proteins [46]. This activation provides broad and long-lasting cytoprotection, enabling cells to adapt and survive under conditions of oxidative, electrophilic, and inflammatory stress [288].

Interestingly, Nrf2 has been found to have dual functions. While it protects normal cells against oxidative stress, cancer cells can hijack Nrf2 activation to promote their survival under unfavorable conditions and develop resistance to various therapies [288]. Moreover, Nrf2’s role extends beyond antioxidant defense, as it has been implicated in regulating metabolism, inflammation, and immunity. In cancer stem cells (CSCs), Nrf2 plays a crucial protective role through multiple mechanisms (Figure 4). It regulates the expression of antioxidant and detoxifying enzymes that defend against oxidative stress and carcinogens [287]. Nrf2 activation induces cytoprotective proteins like NQO1, HO-1, UGT, and GST, which help maintain cellular redox balance and eliminate reactive species that could damage DNA and promote carcinogenesis. The role of Nrf2 in cancer is complex and context-dependent [189]. While generally considered protective, elevated Nrf2 activity may, in some cases, promote tumor growth and chemoresistance [289]. Additionally, NRF2 interacts with other cellular pathways and proteins like RAC3, which can modulate its activity in cancer cells [290].

Nrf2 is crucial in maintaining CSC self-renewal and resistance to chemotherapy and radiotherapy by regulating antioxidant defense mechanisms. Studies have shown that chemotherapy-resistant breast cancer cells exhibit increased Nrf2 stabilization, leading to higher antioxidant enzyme levels and reduced reactive oxygen species (ROS), which supports the CSC phenotype [290]. In glioblastoma, Nrf2 knockdown suppressed proliferation, neurosphere formation, and tumorigenicity [291]. Nrf2 activation is observed in CSC models across various cancers, including lung, esophageal, breast, ovarian, and colon, where it helps maintain low ROS levels and enhances chemoresistance [39]. Nrf2-mediated upregulation of drug efflux transporters like BCRP and MRP2 further contributes to CSC survival [39]. Nrf2 knockdown sensitized CSCs to chemotherapy by impairing antioxidant and drug resistance pathways, underscoring its significance in cancer stem cell biology and therapeutic resistance [292].

Beyond its role in cancer, Nrf2 is crucial for cellular defense mechanisms, particularly in regulating mitochondrial function and maintaining mitochondrial integrity [293]. It counteracts ROS production and supports mechanisms like mitophagy and resistance to permeability transition pore opening. The importance of Nrf2 in mitochondrial health is evident from its deficiency being linked to mitochondrial dysfunction in neurodegenerative disorders such as Alzheimer’s, Parkinson’s, and Friedreich’s ataxia (FRDA) [294]. Conversely, Nrf2 activation enhances mitochondrial biogenesis, fatty acid oxidation, and ATP production. Nrf2 signaling also plays a significant role in modulating neuroinflammation [295]. It suppresses proinflammatory cytokines in microglia and astrocytes, with Nrf2 activators like sulforaphane and methysticin demonstrating the ability to reduce neuroinflammation and oxidative damage in disease models. Furthermore, Nrf2 is essential for autophagy regulation, facilitating the clearance of misfolded proteins and damaged mitochondria [296]. Its activation enhances the expression of autophagy-related genes, contributing to cellular homeostasis. The intricate interplay between Nrf2, mitochondrial function, neuroinflammation, and autophagy underscores its therapeutic potential, particularly in neurodegenerative diseases. This multifaceted role of Nrf2 in both cancer and neurodegenerative disorders highlights its significance as a key regulator of cellular health and stress response, making it an important target for therapeutic interventions in various pathological conditions.

#### 5.1.10. Oxidative Stress and Metabolic Reprogramming

Cancer stem cells (CSCs) and neurodegenerative diseases both involve significant metabolic reprogramming but differ in their mechanisms and consequences. CSCs exhibit metabolic flexibility, switching between oxidative phosphorylation (OXPHOS) and glycolysis, allowing them to adapt, proliferate, and survive under various conditions, including hypoxia [297]. This flexibility is accompanied by enhanced pentose phosphate pathway (PPP) activity, crucial for maintaining stemness and drug resistance [298].

CSCs actively utilize reactive oxygen species (ROS) for signaling purposes. Low ROS levels act as signaling messengers, promoting cancer cell proliferation and invasion, while excessive ROS can induce apoptosis or tumor metastasis [299]. This fine-tuned balance of ROS levels supports CSC survival and function.

In contrast, neurodegenerative diseases often exhibit a shift toward glycolysis as a compensatory response to mitochondrial dysfunction. Microglial cells, for example, reprogram metabolically, transitioning from OXPHOS to glycolysis, characterized by increased glucose uptake, lactate production, and upregulation of glycolytic enzymes [300]. Aging brains similarly show reduced glycolysis, mitochondrial dysfunction, and heightened ROS generation in the electron transport chain, leading to oxidative stress [300].

Interestingly, this metabolic shift is not unique to neurodegeneration. For instance, cancer cells with activated K-ras(G12V) mutations show mitochondrial dysfunction, decreased respiration, and elevated glycolysis [301]. Similarly, senescent cells accumulate dysfunctional mitochondria, which produce excessive ROS, leading to impaired OXPHOS and increased reliance on glycolysis [302].

Metabolic adaptation plays a crucial role in maintaining redox homeostasis in cancer stem cells (CSCs) and neurodegenerative conditions. In CSCs, mitochondrial redox metabolism is central to cellular metabolic reprogramming, allowing these cells to adapt to various stages of cancer progression. The flexibility of mitochondrial metabolism in CSCs is pivotal for their adaptation to new environments during metastasis, with oxidative metabolism being suppressed to lower reactive oxygen species (ROS) generation and maintain optimal NADPH levels for redox homeostasis. Interestingly, while cancer cells generally exhibit increased ROS levels, some CSCs maintain low ROS levels, similar to normal stem cells [67]. This unique redox pattern in CSCs is partly regulated by the NRF2-KEAP1 system, which orchestrates a pleiotropic response to maintain redox homeostasis and confer adaptation to stress conditions [303]. The expression level of NRF2 is higher in CSCs, indicating a stronger dependence on this transcription factor for redox balance. In neurodegenerative conditions, maintaining redox homeostasis is essential for brain cell survival due to their high metabolic energy requirements [303]. OS can lead to mitochondrial damage and protein aggregation, contributing to neurodegeneration. Autophagy plays a protective role in neurodegenerative diseases by removing ROS, damaged mitochondria, and aggregated proteins, thus maintaining redox homeostasis in the brain [304]. Additionally, the FoxO3-dependent metabolic program in neural stem/progenitor cells supports redox balance and neurogenic potential by regulating enzymes in central carbon metabolism [304]. In conclusion, metabolic adaptation significantly influences redox homeostasis in both CSCs and neurodegenerative conditions. Understanding these mechanisms can provide insights into potential therapeutic targets for cancer treatment and neuroprotection. For CSCs, targeting NRF2 or inducing ferroptosis by disrupting iron/lipid metabolism may be promising strategies. In neurodegenerative diseases, promoting autophagy through pharmacological intervention or genetic activation could be an effective approach for maintaining redox balance and protecting against neurodegeneration.

In conclusion, while CSCs leverage glycolysis and ROS signaling for survival and proliferation, neurodegenerative diseases rely on glycolysis as a compensatory mechanism for mitochondrial impairment. These insights into metabolic reprogramming and oxidative stress provide avenues for developing targeted therapies for cancer and neurodegenerative diseases.

#### 5.1.11. Oxidative Stress: Inverse Relationships in Cancer and NDDs

The central nervous system’s high metabolic activity and relatively low antioxidative capacity make it particularly susceptible to oxidative damage, potentially explaining the prevalence of neurodegenerative diseases. In both cancer and neurodegenerative diseases, mitochondrial dysfunction has been identified as an early event, leading to increased ROS production and oxidative stress [305]. This shared mechanism suggests that strategies aimed at improving mitochondrial function or enhancing ROS scavenging may have potential clinical relevance for both types of diseases.

The inverse relationship between cancer and neurodegenerative diseases (NDDs) suggests a complex interplay of shared biological mechanisms that are dysregulated in opposite directions [306]. Several genes and pathways are implicated in both conditions (Table 4):PARK2 (Parkin) and PARK5 (UCHL1): Associated with Parkinson’s disease (PD) and have roles in cancer and oxidative stress. Parkin is involved in mitochondrial quality control and protects against oxidative stress-induced neurodegeneration [307]. In cancer, PARK2 acts as a tumor suppressor. UCHL1, part of the ubiquitin-proteasome system, is downregulated in some cancers but upregulated in others [308].APOE: Known for its role in Alzheimer’s disease (AD), it also influences oxidative stress and cancer risk. The APOE ε4 allele is associated with increased oxidative stress and higher AD risk while potentially offering protection against certain cancers [309,310].PTEN: A tumor suppressor gene that regulates the PI3K/AKT pathway involved in cell survival and proliferation [311]. In NDDs, PTEN dysfunction can lead to increased oxidative stress and neuronal death, while in cancer, PTEN loss promotes cell survival and tumor growth [312].DJ-1: A redox-responsive cytoprotective protein involved in regulating oxidative stress and linked to Parkinson’s disease (PD). It acts as a transcriptional regulator of antioxidative genes and controls oxidative stress in ischemia, neuroinflammation, and age-related neurodegenerative processes [313]. DJ-1 is also connected to Nrf2, a master regulator of antioxidant gene expression. DJ-1 is involved in hepatocellular carcinoma (HCC) development, with a significant inverse correlation between DJ-1 expression and overall survival in HCC patients [314]. DJ-1 knockout mice displayed reduced tumorigenesis and cell proliferation, accompanied by decreased hepatic inflammation and IL-6/STAT3 activation in a DEN-induced murine HCC model [314].ABCA7: An ATP-binding cassette transporter identified as a susceptibility factor for late-onset Alzheimer’s disease (AD) [315]. It plays a role in amyloid precursor protein (APP) processing and amyloid-β (Aβ) generation. Loss of ABCA7 function results in increased β-secretase cleavage and elevated Aβ levels [316]. ABCA7 also mediates phagocytosis and affects membrane trafficking. In retinoblastoma, the Y79 cell line demonstrates high gene expression of ABCA7 along with several other ABC transporters [317]. This elevated expression suggests that ABCA7 might be a potential target for medical treatment of retinoblastoma.MAPT (Tau): A microtubule-associated protein implicated in various neurodegenerative diseases. Recent evidence suggests its involvement in DNA repair and p53 regulation, indicating a potential role in cancer. MAPT expression is associated with key cancer hallmarks and clinical outcomes in a context-specific manner [318]. The involvement of these genes in both cancer and NDDs, often with opposing effects, underscores the complex relationship between oxidative stress, cell survival, and disease pathogenesis. An understanding of these shared pathways may lead to novel therapeutic approaches for both cancer and neurodegenerative disorders.

**Table 4 cells-14-00511-t004:** Oxidative biomarkers of CSCs and NDDs.

Biomarker	Description	Relevance to Cancer	Relevance to NDDs
F2-Isoprostanes	Lipid peroxidation products formed by free radical attack on arachidonic acid.	Elevated in various cancers due to increased oxidative stress and inflammation [319].	Biomarker for lipid peroxidation in Alzheimer’s and Parkinson’s disease; associated with cognitive decline [320].
8-Oxo-2′-deoxyguanosine (8-oxo-dG)	Oxidative DNA damage marker resulting from reactive oxygen species (ROS) attack on guanine bases.	High levels found in tumor tissues, indicating genomic instability and cancer progression [321].	Increased in Alzheimer’s, Parkinson’s, and ALS; contributes to neuronal DNA damage and apoptosis [322].
Glutathione (GSH)/Glutathione Disulfide (GSSG) Ratio	Indicator of cellular redox status; higher GSH/GSSG suggests better antioxidant defense.	Reduced in tumor tissues, indicating oxidative stress and impaired detoxification [323].	Low GSH/GSSG ratio in neurodegenerative diseases suggests oxidative stress-induced neuronal damage [324].
Malondialdehyde (MDA)	Byproduct of lipid peroxidation; marker of oxidative damage to membranes.	Elevated in cancer patients, linked to tumor growth and progression [325].	increased in AD and PD; contributes to neuronal degeneration [326].
Protein Carbonyls	Markers of protein oxidation and dysfunction.	Increased levels found in various cancers, reflecting oxidative damage to proteins [327].	Elevated in Alzheimer’s, Parkinson’s, and Huntington’s disease; associated with neurotoxicity [328].
SOD, CAT, and Glutathione Peroxidase (GPx)	Enzymatic antioxidants that neutralize ROS.	Altered expression in cancer; reduced activity may promote tumor progression [329].	Reduced levels contribute to oxidative stress in neurodegenerative diseases [330].
Nitrotyrosine	Marker of peroxynitrite-mediated nitrosative stress.	High levels linked to cancer development and inflammation [331].	Increased in Alzheimer’s and Parkinson’s; associated with mitochondrial dysfunction [332].

## 6. Future Directions, Challenges, and Gaps in Current Knowledge

OS and redox imbalance play crucial roles in both CSCs and NDDs, presenting common mechanisms and challenges in understanding their pathophysiology. The overproduction of ROS and RNS can lead to oxidative tissue damage, which is a critical event in neurodegenerative diseases. Similarly, in cancer, ROS can function as a double-edged sword, modulating various aspects of cancer development and survival. One of the main challenges in studying oxidative stress is the complex interplay between ROS production and antioxidant defense mechanisms. While controlled oxidative metabolism and redox signaling are crucial for maintaining brain function, excessive ROS production can damage neurons. This delicate balance is also observed in stem cells, where the redox state regulates the balance between self-renewal and differentiation. The dual nature of ROS in both physiological and pathological conditions makes it difficult to develop effective therapeutic strategies.

Despite the growing evidence linking oxidative stress to neurodegenerative diseases and cancer, there are significant gaps in our current knowledge. Antioxidant therapies have shown inconsistent results in treating neurodegenerative diseases, and human experience with antioxidant neuroprotectants has generally been negative in terms of clinical progress. Similarly, in cancer research, prevention by antioxidants has been mostly inefficient. While antioxidants have demonstrated potential in scavenging free radicals and preventing oxidative damage, their clinical efficacy has been disappointing [333]. This is primarily attributed to poor bioavailability and inefficient delivery to the central nervous system (CNS) due to blood–brain barrier limitations [334]. One of the main challenges is the complex, multifactorial nature of NDDs and cancer. These disorders involve various pathways beyond just oxidative stress, including genetic factors, environmental risks, and mitochondrial dysfunction [335]. Additionally, the timing and dosage of antioxidant interventions are critical, as oxidative stress can vary throughout disease progression. Despite their potential, conventional antioxidant therapies face several obstacles. Poor solubility, inefficient permeability, instability during storage, first-pass effect, and gastrointestinal degradation all contribute to their limited efficacy. In order to overcome these limitations, novel drug delivery systems and nanoengineered platforms are being explored to improve bioavailability and targeted delivery of antioxidants to the CNS. Furthermore, research is shifting toward more comprehensive approaches, such as targeting mitochondrial dysfunction and epigenetic mechanisms, which may offer more promising avenues for treating NDDs and cancer.

These challenges necessitate a more nuanced perception of redox biology and the development of targeted approaches to modulate oxidative stress in both cancer and neurodegenerative diseases.

Future research directions should focus on elucidating the complex role of oxidative stress in cancer and neurodegenerative diseases, with the aim of developing more effective therapeutic strategies. Key areas of investigation include the following:Molecular mechanisms: Unraveling the specific pathways through which reactive oxygen species (ROS) and reactive nitrogen species (RNS) contribute to the pathogenesis of neurodegenerative diseases and cancer.Mitochondrial dysfunction: Examining the relationship between mitochondrial impairment, oxidative stress, and their impact on cellular processes in both conditions.Biomarker development: Creating more sensitive and specific indicators of oxidative stress to enhance early detection and disease progression monitoring.Targeted antioxidant therapies: Exploring interventions that can selectively modulate ROS levels in affected tissues without disrupting physiological redox signaling.Cellular process interactions: Investigating the interplay between oxidative stress and other cellular mechanisms, such as inflammation, autophagy, and apoptosis, in the context of neurodegenerative diseases and cancer.Epigenetic modifications: Studying the role of oxidative stress-induced epigenetic changes in disease progression and potential therapeutic interventions.Lifestyle factors: Examining the impact of diet, exercise, and other lifestyle elements on redox balance and their potential as complementary approaches to manage oxidative stress-related diseases.Combination therapies: Investigating treatments that target multiple aspects of redox imbalance and oxidative stress-induced damage.Advanced drug delivery: Exploring nanotechnology and innovative drug delivery systems in order to enhance the efficacy and specificity of antioxidant therapies.Clinical trials: Conducting comprehensive, long-term studies to evaluate the effectiveness of novel antioxidant strategies in preventing or treating neurodegenerative diseases and cancer.

## 7. Conclusions

The review attempts to explore the complex relationship between oxidative stress, redox imbalance, and their roles in cancer and NDDs. The intricate interplay between ROS production, antioxidant defense mechanisms, and cellular signaling pathways highlights the importance of redox homeostasis in maintaining cellular health. In cancer, the dual nature of ROS as both promoters and suppressors of tumor growth underscores the need for a nuanced understanding of redox balance in different cancer types and stages. Similarly, in NDDs, the accumulation of oxidative damage and dysregulation of redox-sensitive pathways contribute to neuronal death and disease progression. The shared mechanisms of oxidative stress and redox imbalance in cancer and NDDs present both challenges and opportunities for therapeutic interventions. Future research should focus on developing targeted approaches for modulating redox balance, considering the specific cellular contexts and disease stages. Such strategies may lead to more effective treatments for both cancer and NDDs, potentially offering new avenues for the prevention and management of these complex disorders.

## Figures and Tables

**Figure 1 cells-14-00511-f001:**
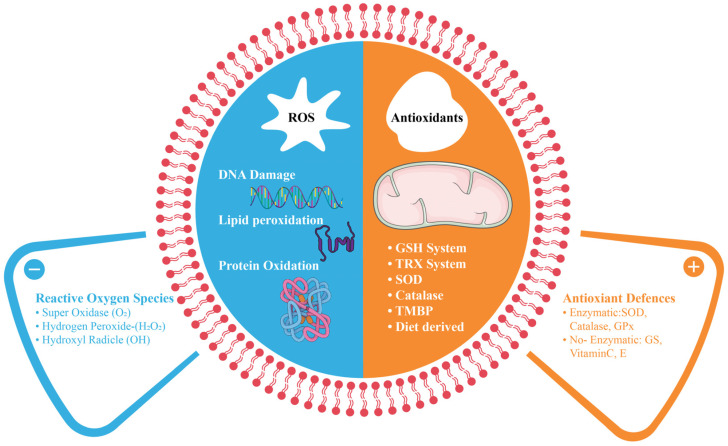
The composition of the antioxidant system and oxidative stress. The core antioxidant systems include glutathione and thioredoxin system (GSH/TRX), superoxide dismutase (SOD), catalase (CAT), transition metal ion binding proteins (TMBP), and diet-derived antioxidants. These work together to prevent ROS-mediated oxidative damage by preventing protein carbonylation, lipid peroxidation, and DNA damage.

**Figure 2 cells-14-00511-f002:**
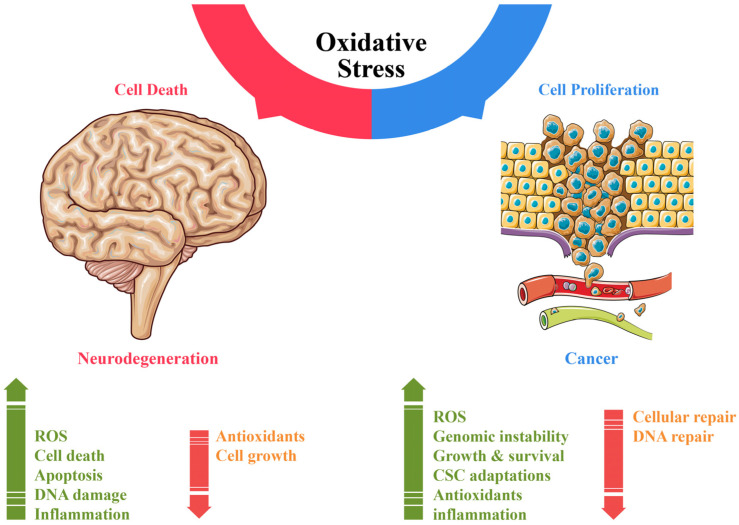
An overview of oxidative stress-mediated overlapping mechanisms of neurodegeneration and cancer.

**Figure 3 cells-14-00511-f003:**
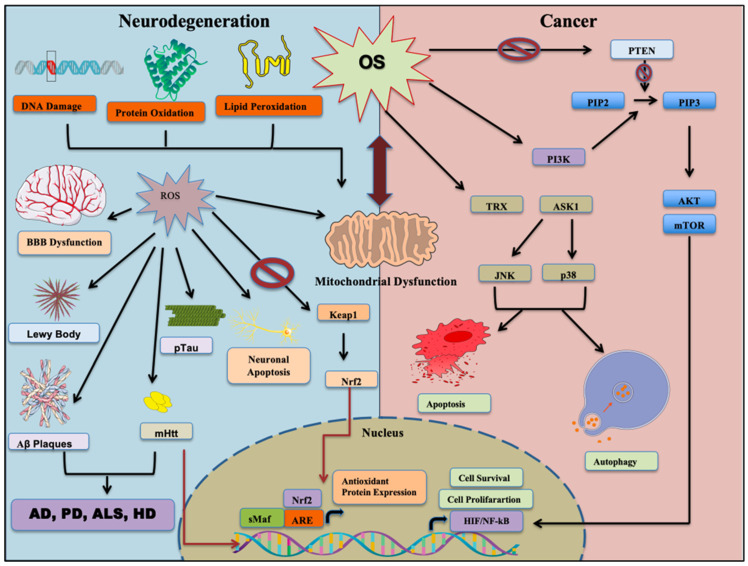
Schematic representing the effect of oxidative stress in neurodegenerative diseases and cancer. An imbalance between ROS and antioxidants results in oxidative stress, which damages essential cellular components such as lipids, proteins, and DNA. Mitochondrial dysfunction, along with microglial activation, triggers the release of inflammatory cytokines and chemokines, ultimately leading to cell apoptosis and tissue degeneration. During oxidative stress, Keap1 is inhibited, and Nrf2 translocates into nucleus, upregulating ARE genes (antioxidant response elements) to counteract OS. Reactive oxygen species (ROS) play a crucial role in regulating cancer-related signaling pathways. An increase in ROS inhibits PTEN, leading to sustained activation of the PI3K/Akt pathway, which promotes cell survival and proliferation. In parallel, oxidative stress triggers the release of ASK-1 from the Trx-ASK1 complex, activating downstream p38 and JNK signaling, which can induce apoptosis and autophagy. Additionally, ROS-mediated inactivation of PP2A supports persistent NF-κB signaling, driving processes such as cell migration, invasion, and uncontrolled growth. These interconnected pathways highlight ROS as a key regulator of both cancer progression and neurodegeneration. Nrf2: Nuclear factor erythroid 2-related factor 2; JNK: c-Jun N-terminal kinase; Keap1: Kelch-like ECH-associated protein 1; mTOR: mammalian target of rapamycin; PI3K: phosphoinositide 3-kinase; PIP: phosphatidylinositol phosphate; PTEN: phosphatase and tensin homolog; ASK1: Apoptosis Signal-Regulating Kinase 1; TRX: thioredoxin; HIF: hypoxia-inducible facto; NF-κB: nuclear factor-kappa B.

**Figure 4 cells-14-00511-f004:**
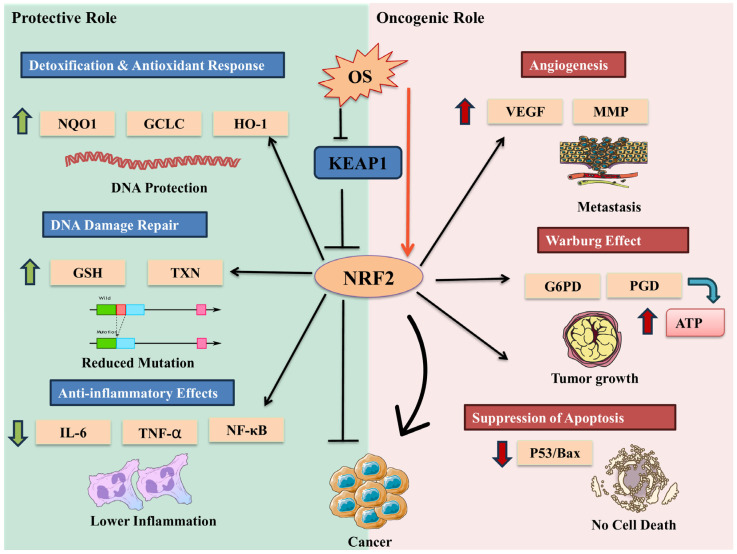
Contradictory roles of Nrf2 in cancer: The contradictory role of Nrf2 in cancer is evident through its diverse functions. On one hand, Nrf2 exhibits protective properties by activating DNA repair mechanisms, regulating antioxidant enzymes, and suppressing pro-inflammatory pathways. These actions potentially reduce cancer-causing mutations, protect cells from oxidative stress, and mitigate chronic inflammation associated with cancer development. On the other hand, Nrf2 demonstrates oncogenic functions by promoting angiogenesis, enhancing metabolic reprogramming in cancer cells, suppressing apoptosis, and potentially promoting tumor growth and chemoresistance when constitutively activated. This dual nature underscores the complexity of Nrf2’s role in cancer biology.

**Table 2 cells-14-00511-t002:** Comparative analysis of key features in cancer stem cells (CSCs) and neurodegenerative diseases (NDDs).

Features	CSCs	NDDs
Cellular Behavior	Uncontrolled proliferation and tumorigenesis [184]	Progressive loss of neurons and synapses [185]
ROS levels and Antioxidant defenses	Generally maintain lower ROS levels compared to non-CSCs; Enhanced antioxidant systems (e.g., increased SOD2, GSH) [186]	Exhibit elevated ROS levels; Impaired antioxidant responses [187]
Mitochondrial function	Metabolic flexibility; can switch between OXPHOS and glycolysis [188]	Early mitochondrial dysfunction [189]
Nrf2 pathway	Dual role: promotes survival and chemoresistance and anticancer activity	Attenuation increases oxidative stress [190]
Autophagy	Can promote both stemness maintenance and loss [189]	Impairment contributes to protein aggregation [191]
Genetic influence	Oncogenes promote cell survival and growth (e.g., PARK7/DJ-1 antagonizing PTEN) [192]	Tumor suppressor-like genes involved in neuroprotection (e.g., *PARK2/Parkin*, *PARK5*) [193]
Inflammation	Immune evasion and immunosuppressive microenvironment [194]	Chronic neuroinflammation and microglial activation [195]
Therapeutic Focus	Targeting proliferative signaling, redox balance to eliminate CSCs and immune evasion (e.g., chemotherapy, immunotherapy) [84]	Enhancing neuronal survival and reducing misfolded proteins (e.g., anti-amyloid, neuroprotective agents), and boosting antioxidant defenses [196].

**Table 3 cells-14-00511-t003:** Metabolic reprogramming in CSCs and NDDs.

Metabolic Process	CSC	NDDs	Examples
Glycolysis	Upregulated even in oxygen-rich conditions (Warburg effect) [208]. Supports rapid proliferation and resistance to apoptosis [209].	Decreased glucose uptake and glycolysis [210]. Leads to energy deficits and neuronal dysfunction [210].	CSCs: Breast CSCs show increased HK2 and LDHA expression [211]. Neurons: Reduced GLUT3 in Alzheimer’s disease [212].
Oxidative Phosphorylation (OXPHOS)	Can be flexible; some CSCs rely on OXPHOS for survival under stress [213].	Impaired due to mitochondrial dysfunction, leading to ROS accumulation [214].	CSCs: Glioblastoma CSCs shift between glycolysis and OXPHOS [215]. Neurons: Mitochondrial Complex I defects in Parkinson’s disease [215]
Pentose Phosphate Pathway (PPP)	Enhanced to generate NADPH for antioxidant defense and nucleotide synthesis [41].	Reduced PPP activity, increasing oxidative stress and impairing neuronal repair [216].	CSCs: Leukemia stem cells upregulate G6PD for redox balance [217]. Neurons: Decreased G6PD activity in Alzheimer’s disease [218].
Fatty Acid Oxidation (FAO)	Upregulated in some CSCs to sustain energy production and stemness [219].	Decreased FAO leads to lipid accumulation and neurotoxicity [220].	CSCs: Ovarian CSCs use FAO to maintain survival [221,222]. Neurons: Impaired FAO in Huntington’s disease leads to lipid droplet buildup [222].

## Data Availability

No new data were created or analyzed in this study.

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
