# Peer review of "Oxidative Stress and Redox Imbalance: Common Mechanisms in Cancer Stem Cells and Neurodegenerative Diseases"

_cells, 2025, doi:10.3390/cells14070511_

Round 1

Reviewer 1 Report

Comments and Suggestions for Authors

General

Houck and co-authors recently published a review article titled "At the crossroads between neurodegeneration and cancer: a review of overlapping biology and its implications." in the journal “Current aging science” (2018). The central theme of their work overlaps with the current review article, “Oxidative Stress and Redox Imbalance: Common Mechanisms in Cancer Stem Cells and Neurodegenerative Diseases”. However, the contents of the current review fall short of the standards expected of high-impact journals like Cells.

Figure 1 contains notable deficiencies, and some inline citations, as well as references, are incorrectly cited. Moreover, many statements in the review fail to align with the title of the study, creating a disconnect between the content and the intended theme. The numbering of the subheadings is incorrect and out of order. Additionally, the references are not formatted according to the journal’s guidelines.

In my opinion, this review article does not meet the standards required for publication in a prestigious journal like Cells.

Comments

Line 33: The reference cited is incorrect.

Line 43, 99, 106, 227, 239, 255, 284, 434, 495, 507, 531, 776, 819 and Figure 1: The spelling of the word “Defense” varies throughout the manuscript, using both American and British English. Please ensure consistency in the English style used across the entire manuscript.

Line 44: Cite references 6 and 7 together in line 44.

Line 54: The reference cited focuses solely on ovarian cancer cells and does not align with the sentence, “Potential … CSCs or neurons.”

Line 56: The reference provided pertains exclusively to breast cancer and does not support the sentence, “For example, enhancing … to treatment”.

Figure 1:

In the upper-left box, ensure the correct notation for “superoxide (O2-), hydroxyl radical (·OH)” and use appropriate subscripts for “O2” and “H2O2”

In the upper-right box, correct the spelling of “Antioxiant Defences” and adjust the spacing after words. Replace “no- enzymatic” with “Non-enzymatic”

Line 110-112: The sentence does not align with the title of the review article, and the reference cited is incorrect.

Line 114-116: The reference cited is incorrect, as the paper does not discuss “cardiomyocytes”

Line 116-117: The sentence, “Additionally, OS influences cell cycle dysregulation and proliferation, as demonstrated by the effects of 25-hydroxycholesterol on extravillous trophoblasts [33],” is peripheral to the article's focus on CSCs and neurodegeneration. The transition “additionally” does not logically follow the previous sentence, “Oxidative stress-induced ferroptosis in cardiomyocytes involves complex mechanisms including glutathione depletion and GPX4 degradation [32]”.

Line 118-120: Please establish a clear connection between pathophysiology of Down syndrome to OS and NDDs. The role of OS in Williams-Beuren syndrome is less documented, and reference “33” focuses only on Down syndrome. Relevant reference for Williams-Beuren syndrome should be incorporated.

Line 315: Correct the heading number

Line 768: Correct the heading number

Line 816: Correct the heading number

Author Response

Comment 1: Line 33: The reference cited is incorrect.

Response 1: Thank you for pointing this out. We have corrected it.

Comment 2: Line 43, 99, 106, 227, 239, 255, 284, 434, 495, 507, 531, 776, 819 and Figure 1: The spelling of the word “Defense” varies throughout the manuscript, using both American and British English. Please ensure consistency in the English style used across the entire manuscript.

Response 2: Thank you for pointing it out. It has been changed to ‘defense’.

Comment 3: Line 44: Cite references 6 and 7 together in line 44.

Response 3: We have done accordingly.

Comment 4: Line 54: The reference cited focuses solely on ovarian cancer cells and does not align with the sentence, “Potential … CSCs or neurons.”

Response 4: Thanks for pointing it out. Reference has been incorporated

Comment 5: Line 56: The reference provided pertains exclusively to breast cancer and does not support the sentence, “For example, enhancing … to treatment”.

Response 5: Agree. Appropriate reference has been added.

Comment 6:

Figure 1:

In the upper-left box, ensure the correct notation for “superoxide (O2-), hydroxyl radical (·OH)” and use appropriate subscripts for “O2” and “H2O2”

In the upper-right box, correct the spelling of “Antioxiant Defences” and adjust the spacing after words. Replace “no- enzymatic” with “Non-enzymatic”

Response 6: Thanks. We have changed and modified the figure. Hope this is fine now.

Comment 7: Line 110-112: The sentence does not align with the title of the review article, and the reference cited is incorrect.

Response 7: Thanks for pointing it out. It has been changed.

Added: OS can cause damage to major biomolecules, including lipids, proteins, carbohydrates, and nucleic acids. This damage can lead to cellular dysfunction, premature cell death, and inflammation. Hope this covers the issue.

Comment 8: Line 114-116: The reference cited is incorrect, as the paper does not discuss “cardiomyocytes”

Response 8: Thanks for pointing it out. Added the reference.

Comment 9: Line 116-117: The sentence, “Additionally, OS influences cell cycle dysregulation and proliferation, as demonstrated by the effects of 25-hydroxycholesterol on extravillous trophoblasts [33],” is peripheral to the article's focus on CSCs and neurodegeneration. The transition “additionally” does not logically follow the previous sentence, “Oxidative stress-induced ferroptosis in cardiomyocytes involves complex mechanisms including glutathione depletion and GPX4 degradation [32]”.

Response 9: Thank you. The section on consequences of oxidative stress was just meant to describe role of oxidative stress, in general, henceforth included cardiomyocytes and trophoblasts.

Comment 10: Line 118-120: Please establish a clear connection between pathophysiology of Down syndrome to OS and NDDs. The role of OS in Williams-Beuren syndrome is less documented, and reference “33” focuses only on Down syndrome. Relevant reference for Williams-Beuren syndrome should be incorporated.

Response 10: Noted. Added the appropriate reference for Down’s, Williams-Beuren connecting NDDs and oxidative stress

Comment 11: Line 315: Correct the heading number

Response 11: Thank you for pointing it out. It has been changed

Comment 12: Line 768: Correct the heading number

Response 12: Thank you for pointing it out. It has been changed

Comment 13: Line 816: Correct the heading number

Response 13: Thank you for pointing it out. It has been changed

Reviewer 2 Report

Comments and Suggestions for Authors

The review is challenging and of importance but very difficult to follow the information. 

Suggestions:

1. to subdivide each section like have been done for section 5

2. once CSC was defined as cancer stem cells 

no need to repeat it. just use only CSC. 

3. figure 2 -I t is not clear how i can learn the overlapping between CSC and NDD

4. raw 315-should be 4 and not 3

5. the importance and uniqueness of the review is the comparison between CSC and not cancer cells with NDD. unfortunately, the authors sometimes give examples from cancer cells -this is misleading (e.g. raw: 139-140, 164, 285-286. and many more). 

Author Response

Comment 1: to subdivide each section like have been done for section 5

Response 1: Thank you. Wherever possible, sections have been subdivided. Hope this is okay.

Comment 2: once CSC was defined as cancer stem cells 

no need to repeat it. just use only CSC. 

Response 2: Thanks for pointing it out. It has been changed accordingly.

Comment 3. figure 2 -I t is not clear how i can learn the overlapping between CSC and NDD

Response 3: Thanks for pointing it out. The figure has been modified, along with addition of new figures.

Comment 4: raw 315-should be 4 and not 3

Response 4: Thanks for pointing it out. It has been changed.

Comment 5. the importance and uniqueness of the review is the comparison between CSC and not cancer cells with NDD. unfortunately, the authors sometimes give examples from cancer cells -this is misleading (e.g. raw: 139-140, 164, 285-286. and many more). 

Response 5: Thanks for pointing it out. It has been changed wherever possible.

Reviewer 3 Report

Comments and Suggestions for Authors

General Assessment

The manuscript provides a comprehensive review of oxidative stress (OS) and redox imbalance, highlighting their roles in cancer stem cells (CSCs) and neurodegenerative diseases (NDDs). The topic is highly relevant, and the manuscript is well-organized, making it accessible to researchers in both oncology and neuroscience. However, there are several critical areas that require improvement, including scientific depth, clarity, citations, and novelty. The manuscript is well-structured and informative, but it requires significant refinements in scientific depth, comparative analysis, therapeutic discussions, and citation support before publication.

Major Concerns

1. The review lacks a clear articulation of how it advances current knowledge. Emphasize what new insights this review offers compared to previous literature.

2. The manuscript discusses CSCs and NDDs separately in many sections instead of making clear direct comparisons. Consider a side-by-side comparative table summarizing similarities and differences.

3. ROS is typically described as harmful, but low ROS levels have beneficial effects in cell signaling. Include a section on "Dual Roles of Oxidative Stress", considering adaptive responses.

4. The NRF2 pathway is mentioned, but contradictory roles in cancer (protective vs. tumor-promoting) and NDDs (neuroprotective vs. dysfunctional) are not well analyzed. Expand discussion on how NRF2 can paradoxically contribute to both tumorigenesis and neurodegeneration.

5. The manuscript suggests NRF2 activators, antioxidants, and NADPH oxidase inhibitors as treatments but fails to discuss why most have failed in clinical trials. Provide an analysis of why antioxidant therapies have been ineffective in treating NDDs or cancer.

6. The role of ferroptosis in both CSCs and neurodegeneration is underdeveloped. Discuss how lipid peroxidation and iron metabolism regulation differ between CSCs and neurons.

7. While mitochondrial ROS is mentioned, there is limited discussion on mitochondrial biogenesis, mitophagy, and metabolic shifts in CSCs vs. neurons. Expand the mitochondrial dysfunction section with examples of metabolic reprogramming in CSCs and degenerating neurons.

8. Aging is briefly mentioned but not fully explored as a major driver of redox dysregulation in both CSCs and NDDs. Add a subsection discussing aging-related oxidative damage.

9. The manuscript mentions NF-κB, MAPK, and KEAP1-NRF2 pathways but does not explore their context-dependent roles in cancer vs. neurodegeneration. Provide a pathway-specific breakdown of how redox signaling differs in CSCs and neurons.

10. OS-driven epigenetic modifications (e.g., DNA methylation, histone acetylation) are absent from the discussion. Add a section on epigenetic consequences of oxidative stress in cancer and NDDs.

11. The Warburg effect in CSCs and glycolysis shifts in neurodegeneration are briefly mentioned but not explored in depth. Expand on how metabolic adaptation affects redox homeostasis in both conditions.

12. Lipid peroxidation is critical in ferroptosis but not discussed in detail. Explain how PUFAs (polyunsaturated fatty acids) modulate oxidative damage in CSCs and neurons.

13. The review suggests ROS kills cancer cells, but CSCs use ROS scavenging for drug resistance. Discuss how CSCs exploit redox regulation to survive chemotherapy and radiation therapy.

14. Iron accumulation is linked to both ferroptosis (cancer) and neurodegeneration, yet this connection is not well developed. Discuss the dual role of iron in oxidative stress-driven damage.

15. Excessive antioxidant use can disrupt redox signaling and promote cancer cell survival. Include why indiscriminate antioxidant use may be harmful.

16. Figures are referenced but lack detailed explanations or clear visualizations of pathways. Provide a figure comparing oxidative stress pathways in CSCs vs. neurons.

17. The manuscript largely discusses antioxidants, but emerging therapies target oxidative metabolism, proteostasis, and mitochondrial dynamics. Expand therapeutic approaches beyond direct ROS scavenging.

18. Different cancers have different ROS levels and antioxidant defenses, which is not addressed. Clarify that not all CSCs rely on low ROS; some use oxidative metabolism to resist therapy.

19. Clinical biomarkers of oxidative damage (e.g., F2-isoprostanes, 8-oxo-dG, GSH/GSSG ratio) are not mentioned. Include a table summarizing oxidative stress biomarkers relevant to cancer and NDDs.

20. Some bold claims lack direct references, particularly in discussions of therapeutic potential. Ensure all mechanistic statements are backed by primary research.

Minor Issues

21. Some sentences are overly complex and should be simplified for clarity.

22. Ensure all acronyms (e.g., NRF2, GSH, RNS) are clearly defined upon first mention.

23. Use oxidative stress, redox imbalance, ROS regulation consistently to avoid redundancy.

Comments on the Quality of English Language

The manuscript is generally well-written and demonstrates a good command of scientific terminology, but there are several areas where language refinement is needed for clarity, readability, and coherence.

Author Response

Comment 1. The review lacks a clear articulation of how it advances current knowledge. Emphasize what new insights this review offers compared to previous literature.

Response 1: Thank you. Lines 60-75, have added to emphasize the insights that this review offers

Comment 2. The manuscript discusses CSCs and NDDs separately in many sections instead of making clear direct comparisons. Consider a side-by-side comparative table summarizing similarities and differences.

Response 2: Thank you for pointing it out. It has been incorporated.

Comment 3. ROS is typically described as harmful, but low ROS levels have beneficial effects in cell signaling. Include a section on "Dual Roles of Oxidative Stress", considering adaptive responses.

Response 3: Thank you for pointing it out. A new section on Dual roles of OS has been added.

  1. The NRF2 pathway is mentioned, but contradictory roles in cancer (protective vs. tumor-promoting) and NDDs (neuroprotective vs. dysfunctional) are not well analyzed. Expand discussion on how NRF2 can paradoxically contribute to both tumorigenesis and neurodegeneration.

Response 4: Thank you for pointing it out. A new section on Nrf2 has been added, along with a pictorial representation.

  1. The manuscript suggests NRF2 activators, antioxidants, and NADPH oxidase inhibitors as treatments but fails to discuss why most have failed in clinical trials. Provide an analysis of why antioxidant therapies have been ineffective in treating NDDs or cancer.

.

Response 5: Thank you for pointing it out. This has been incorporated in the final section

  1. The role of ferroptosis in both CSCs and neurodegeneration is underdeveloped. Discuss how lipid peroxidation and iron metabolism regulation differ between CSCs and neurons.

Response 6: Thank you for pointing it out. This has been incorporated in section.

  1. While mitochondrial ROS is mentioned, there is limited discussion on mitochondrial biogenesis, mitophagy, and metabolic shifts in CSCs vs. neurons. Expand the mitochondrial dysfunction section with examples of metabolic reprogramming in CSCs and degenerating neurons.

Response 7: A table has been included (table 3)

  1. Aging is briefly mentioned but not fully explored as a major driver of redox dysregulation in both CSCs and NDDs. Add a subsection discussing aging-related oxidative damage.

Response 8: Added as a subsection

  1. The manuscript mentions NF-κB, MAPK, and KEAP1-NRF2 pathways but does not explore their context-dependent roles in cancer vs. neurodegeneration. Provide a pathway-specific breakdown of how redox signaling differs in CSCs and neurons.

Response 9: A table has been included (table 1)

  1. OS-driven epigenetic modifications (e.g., DNA methylation, histone acetylation) are absent from the discussion. Add a section on epigenetic consequences of oxidative stress in cancer and NDDs.

Response 10: Added as a subsection

  1. The Warburg effect in CSCs and glycolysis shifts in neurodegeneration are briefly mentioned but not explored in depth. Expand on how metabolic adaptation affects redox homeostasis in both conditions.

Response 11: brief explanation included

  1. Lipid peroxidation is critical in ferroptosis but not discussed in detail. Explain how PUFAs (polyunsaturated fatty acids) modulate oxidative damage in CSCs and neurons.

Response 12: added as a subsection in ferroptosis and oxidative stress

  1. The review suggests ROS kills cancer cells, but CSCs use ROS scavenging for drug resistance. Discuss how CSCs exploit redox regulation to survive chemotherapy and radiation therapy.

Response 13: briefly described in the conclusion part

  1. Iron accumulation is linked to both ferroptosis (cancer) and neurodegeneration, yet this connection is not well developed. Discuss the dual role of iron in oxidative stress-driven damage.

Response 14: added as a subsection in ferroptosis and oxidative stress

  1. Excessive antioxidant use can disrupt redox signaling and promote cancer cell survival. Include why indiscriminate antioxidant use may be harmful.

Response 15: briefly described in the challenges

  1. Figures are referenced but lack detailed explanations or clear visualizations of pathways. Provide a figure comparing oxidative stress pathways in CSCs vs. neurons.

Response 16: figure 3 included

  1. The manuscript largely discusses antioxidants, but emerging therapies target oxidative metabolism, proteostasis, and mitochondrial dynamics. Expand therapeutic approaches beyond direct ROS scavenging.

Response 17: briefly described in conclusion

  1. Different cancers have different ROS levels and antioxidant defenses, which is not addressed. Clarify that not all CSCs rely on low ROS; some use oxidative metabolism to resist therapy.

Response 18: briefly described the dual roles of oxidative stress and pathways like nrf2 in CSCs

  1. Clinical biomarkers of oxidative damage (e.g., F2-isoprostanes, 8-oxo-dG, GSH/GSSG ratio) are not mentioned. Include a table summarizing oxidative stress biomarkers relevant to cancer and NDDs.

Response 19: Added as a table (table 4)

  1. Some bold claims lack direct references, particularly in discussions of therapeutic potential. Ensure all mechanistic statements are backed by primary research.

Response 20: references has been included in detail.

Round 2

Reviewer 1 Report

Comments and Suggestions for Authors

The authors have made significant revisions to the manuscript and have adequately addressed the comments and suggestions provided in the previous review. The concerns regarding incorrect citations, inconsistencies in terminology, formatting of references and figure deficiencies have been carefully revised. The alignment between the content and the central theme of the review has been improved, ensuring coherence with the title. Additionally the reference formatting has been adjusted according to the journal's guidelines. Based on these improvements, the revised manuscript has been significantly strengthened and now meets the necessary standards for publication.

Author Response

Thank you for the valuable suggestions and helping us restructure the review. 

Thanks

Reviewer 3 Report

Comments and Suggestions for Authors

The author has mostly addressed my concerns in this revised manuscript. However, please include more detailed figure legends for Figures 1 and 2.

Author Response

Thank you for your valuable suggestion and helping us restructure the review. 

Comment: please include more detailed figure legends for Figures 1 and 2.

Reply: Noted. We have made necessary corrections to the legends. Added few additional points.

Thank you